



# The impact of aerosol fluorescence on water vapor long-term monitoring by Raman lidar and the evaluation of a potential correction method

Fernando Chouza[1], Thierry Leblanc[1], Mark Brewer[1], Patrick Wang[1], Giovanni Martucci[2], Alexander Haefele[2], Hélène Vérèmes[3], Valentin Duflot[3], Guillaume Payen[4], Philippe Keckhut[5]

[1]Laboratory Studies and Atmospheric Observations, Jet Propulsion Laboratory, California Institute of Technology, 92397 Wrightwood, USA
[2]Federal Office of Meteorology and Climatology, MeteoSwiss, CH-1530 Payerne, Switzerland
[3]Laboratoire de l'Atmosphère et des Cyclones (LACy, UMR 8105 CNRS, Université de la Réunion, Météo-France), Université de La Réunion, 97400 Saint-Denis de La Réunion, France
[4]Observatoire des Sciences de l'Univers de La Réunion (OSU-Réunion), UAR 3365, Université de la Réunion, CNRS, Météo-France, 97400 Saint-Denis de La Réunion, France
[5]LATMOS/IPSL, UVSQ Université Paris-Saclay, Sorbonne Université, CNRS, 75000 Paris, France

*Correspondence to*: Fernando Chouza (keil@jpl.nasa.gov)

**Abstract.** The impact of aerosol fluorescence on the measurement of water vapor by UV (355 nm emission) Raman lidar in the upper troposphere and lower stratosphere (UTLS) is investigated using the long-term records of three high-performance Raman lidars contributing to the Network for the Detection of Atmospheric Composition Change (NDACC). Comparisons with co-located radiosondes and aerosol backscatter profiles indicate that laser-induced aerosol fluorescence in smoke layers injected into the stratosphere by pyrocumulus events can introduce very large and chronic wet biases above 15 km, thus impacting the ability of these systems to accurately estimate long-term water vapor trends in the UTLS.

In order to mitigate the fluorescence contamination, a correction method based on the addition of an aerosol fluorescence channel was developed and tested on the water vapor Raman lidar TMWAL located at the JPL Table Mountain Facility, in California. The results of this experiment, conducted between 27 August and 4 November 2021 and involving 22 co-located lidar and radiosonde profiles, suggest that the proposed correction method is able to effectively reduced the fluorescence-induced wet bias. After correction, the average difference between the lidar and co-located radiosonde water vapor measurements was reduced to 5%, consistent with the difference observed during periods of negligible aerosol fluorescence interference.

The present results provide confidence that, after a correction is applied, water vapor long-term trends can be reasonably well estimated in the upper troposphere, but they also call for further refinements, or the use of alternate Raman lidar approaches (e.g., 308 nm or 532 nm emission) to confidently detect long-term trends in the lower stratosphere. These findings may have important implications on NDACC's water vapor measurements strategy in the years to come.



# 1 Introduction

Water vapor is a key component of the atmosphere and plays a major role in the Earth's radiative balance. Numerous studies (e.g. Solomon et al., 2010) indicate that small changes in stratospheric water vapor abundance can have a large impact on the
Earth's radiation budget and decadal surface temperature changes, stressing the need for accurate long-term measurements of water vapor (Müller et al., 2016).

Among the different instruments available for the measurement of water vapor in the Upper Troposphere and Lower Stratosphere (UTLS), Raman lidars exhibit a combination of features that make them useful for sustained long-term operation as required for trend detection. These characteristics include a low cost per measured profile, a relatively simple path to
automation, and a good understanding of the physics involved in the measurement which facilitates long-term operation with minimal disturbances. As a result of this apparent sustainability, the Network for the Detection of Atmospheric Composition Change (NDACC) included water vapor Raman lidars among its suite of contributing instruments, with the main goal to detect water vapor trends in the UTLS. Because of the extreme dryness of the UTLS (2-5 ppmv), reaching this layer of the atmosphere requires the Raman lidar instruments to be particularly sensitive and/or powerful.

The detection of long-term trends requires an accurate characterization of the systematic errors affecting the instruments used for such detection, in particular their change with time. If not well characterized, systematic errors in the measurements are likely to interfere with the trend that we are trying to estimate. In the case of water vapor Raman lidars, the systematic errors can have different origins, including changes in the Raman cross-section with temperature (Whiteman, 2003), insufficient Rayleigh and Mie scattering blocking on the Raman channels, calibration (Leblanc et al., 2008; 2012), instrument fluorescence
(Sherlock et al., 1999; Leblanc et al., 2012; Whiteman et al., 2012), aerosol extinction, and aerosol fluorescence (Immler et al., 2005a). Among these, the aerosol fluorescence effect is the most problematic due to the difficulty to characterize and quantify the fluorescence contamination of the water vapor Raman signal given the variability of the fluorescence spectral distribution from an aerosol type to another. While some aerosol types exhibit negligible fluorescing properties in the wavelength region where water vapor lidars typically operate, other types can induce a very large interference (e.g. Veselovskii
et al., 2021).

The first report of aerosol fluorescence contamination on a water vapor Raman lidar measurement was presented by Immler et al. (2005a) and further discussed by Immler et al. (2005b). Since then, several studies have been conducted to use the fluorescence characteristics as an additional source of information for aerosol typing (e.g. Sugimoto et al., 2012; Veselovskii et al., 2021). However, to the best of our knowledge, no thorough study has been conducted to determine the impact of
fluorescence on lidar-based long-term water vapor trend calculations, and no solution has been proposed to this issue besides avoid measuring when fluorescence contamination is suspected. The relative lack of interest in resolving this issue may owe to the fact that fluorescence contamination has remained a rare interference over the years, affecting only the uppermost range of most existing water vapor Raman lidar measurements, which main research objective is focused on meteorology rather than long-term trends. Nevertheless, based on spaceborne aerosol observations by the Cloud-Aerosol Lidar with Orthogonal





Polarization (CALIOP) lidar and the Ozone Mapping and Profiler Suite (OMPS), and based on recent measurements conducted at three NDACC lidar stations, namely the JPL Table Mountain Facility (California, United States), the Meteoswiss station of Payerne (Switzerland), and the Maïdo high-altitude observatory (Reunion Island, France), we found that the fluorescence contamination on the water vapor signals acquired by Raman lidars based on third harmonic Nd:YAG transmitters (355 nm) has become the norm rather than the exception in the UTLS since 2017.

The aim of this work is to assess the impact of fluorescence by aerosols on the long-term water vapor records from these three Raman lidars, as well as to provide a method to correct it, and determine if the accuracy of the correction meets the requirements of long-term water vapor trend studies. Section 2 presents an overview of the instruments and auxiliary datasets used in this work. Section 3 summarizes the suspected impact of aerosol fluorescence on the three NDACC water vapor lidar datasets just mentioned. In section 4 we propose a possible correction method and evaluate its performance. The paper concludes with a
discussion of the results and the implications for long-term water vapor trend analysis based on Raman lidar measurements.

## 2 Instruments and datasets

In this study we include long-term datasets from three NDACC water vapor Raman lidars that use a third harmonic Nd:YAG laser transmitter (355 nm), which is a common choice for water vapor lidar systems aiming to reach the UTLS due to a combination of high Raman scattering efficiency and availability of affordable high power laser sources. Two of these lidar
systems are in the northern hemisphere mid-latitudes, while the other is located in the southern hemisphere sub-tropics. By considering several independent instruments with different designs, we can rule out any specific instrumental issue as the cause of the observed biases. Additionally, since stratospheric aerosols play an important role on this study, and not all the lidar systems involved have aerosol characterization capabilities in the UTLS, we decided to use the Cloud-Aerosol Lidar with Orthogonal Polarization (CALIOP) stratospheric aerosol products extracted at each lidar station as the source of aerosol
distribution information. Finally, when available, we included reference water vapor profiles from co-located radiosondes to help determine the magnitude of the fluorescence-induced bias in the water vapor lidar profiles.

### 2.1 Table Mountain Water Vapor Lidar (TMWAL)

The JPL Table Mountain Water Vapor Lidar (TMWAL) is a high-performance water vapor Raman lidar that started its routine operations in 2005. The system is located at the JPL Table Mountain Facility near Wrightwood, California (34.38° N, 117.68°
W; 2285 m a.s.l.). The most important characteristics of the systems are presented in the next paragraph for the convenience of the reader. A detailed description of the system can be found in Leblanc et al., 2012, while the results of the validation studies conducted during three Measurement Of Humidity And Validation Experiment (MOHAVE) campaigns can be found in Leblanc et al., 2011. The transmitter is based on a high pulse energy (650 mJ per pulse) Nd:YAG laser (manufactured by Continuum) operating at 355 nm with a repetition rate of 10 Hz. To reduce beam divergence and as well as to increase eye
safety, the output laser beam is expanded 7.5 times by a refractive beam expander before being reflected to the atmosphere by



a motorized mirror used for automatic alignment. The main receiver is a 91 cm-diameter Newtonian telescope complemented by four small (75 mm) telescopes for the near range. During the first years of operation and until 2009, TMWAL underwent a series of modifications with the aim to reduce the instrument bias caused by fluorescence generated in the optical fiber used to couple the primary mirror with the receiver polychromator. After switching to a free space coupling approach in 2009, the

instrumental fluorescence was permanently suppressed and the system remained almost unchanged for the following ten years. In 2019, the system underwent minor modifications to allow automated measurements in a similar fashion as the rest of the lidars operated at JPL-TMF (Chouza et al., 2019), changes that did not affect the water vapor measurement characteristics. The measurement schedule of TMWAL typically implies 4 or 5 measurements per week, nighttime only, with each measurement period corresponding to an effective acquisition time of 2-3 hours. The results are archived at the NDACC data

handling center on a quasi-monthly basis. The primary calibration source is co-located radiosonde. As part of the hybrid calibration approach described in Leblanc et al. (2008; 2012), three to five radiosonde launches per month are typically conducted. These launches are normally conducted around new moon and clear skies in order to maximize the lidar signal-to-noise ratio and thus the accuracy of the calibration. As part of the hybrid calibration, a NIST-calibrated lamp is used just before and after each measurement period to monitor the changes in the lidar receiver's transmission from night to night between two

periods of absolute calibration from radiosonde. Thanks to the large receiver, laser power, and elevation of JPL-TMF, the water vapor profiles can reach up to 20 km a.s.l. during new moon (low sky background) and high humidity, while under less favorable conditions (full moon and low humidity) the profiles maximum altitude is about 15 km a.s.l..

**2.2 Raman Lidar for Meteorological Observations (RALMO)**

RALMO was designed and built by the École Polytechnique Fédérale de Lausanne (EPFL) in collaboration with MeteoSwiss.

RALMO is dedicated to operational meteorology, model validation and climatological studies; it also serves as reference ground-based measurement for satellite validation and calibration studies. After its installation at the MeteoSwiss station of Payerne (46.313 N, 6.943 E; 491 m a.s.l.) in 2007, RALMO has provided profiles of humidity, temperature and aerosol backscatter in the troposphere and lower stratosphere almost uninterruptedly since 2008 (Brocard et al., 2013; Dinoev et al., 2013, Martucci et al 2021). RALMO has been designed to achieve a measurement precision better than 10% for humidity and

0.5 K for temperature at high temporal resolution (30-minute integration time). RALMO uses high-energy emission, narrow field of view of the receiver and a narrowband detection to achieve high performance and data quality. RALMO's frequency-tripled Nd:YAG laser emits 450 mJ per pulse at 30 Hz and at 355 nm. A beam expander expands the beam's diameter to 14 cm and reduces the beam divergence to 0.09±0.02 mrad. The returned signal is an envelope of the 355 nm elastic- and Raman-backscattered signals, i.e. pure rotational Raman, water vapor, oxygen, nitrogen and Rayleigh. The receiver's telescope consists

of four 30-cm mirrors, fiber-coupled to a polychromator based on a holographic diffraction grating (3600 mm−1, 85 × 85 mm2). The grating-based polychromator is used instead of interference-filter-based to achieve long-term data consistency and to minimize the temperature dependency. The data acquisition software ensures autonomous operation of the system and real-time data availability. The data used for this study are the water vapor mixing ratio (WVMR) timeseries from 2008 to 2021



and available publicly at the NDACC database. The WVMR profile are integrated in time over the entire night, from an hour

after the astronomical sunset to an hour before the astronomical sunrise. Only profiles with signal-to-noise ratio (SNR) larger than one in the region 0-12 km are retained and submitted to the NDACC database. The condition imposed to the SNR ensures that the data have good quality at least up to the tropopause, assuming good atmospheric conditions and a correct calibration of the humidity channel. RALMO WVMR profiles shall then reach at least 12 km in clear-sky conditions and in wintertime when the integration time is longer.

The data used for the analysis and shown in Fig. 2 are monthly averaged and have been filtered using a relative error threshold of 700% between 10 km and 20 km. The relative error is the combination of the systematic and random error. We apply Gaussian error propagation through the WVMR equation and account for the main sources of uncertainty, which are measurement noise and the calibration coefficient. Measurement noise and atmospheric variability are separated using the temporal autocorrelation function.

**2.3 Lidar1200**

The Lidar1200 is a water vapor Raman lidar designed to measure the water vapor mixing ratio in the troposphere and the lower stratosphere for long-term monitoring and process studies up to the UTLS. The system is located at the Maïdo high-altitude observatory in Reunion Island (21.08° S, 55.38° E ; 2154 m a.s.l.) (Baray et al., 2013). The instrument has been in routine operation and the measurements have been calibrated since November 2013. The design has been based on the preliminary

developments performed in Observatory of Haute-Provence and In La Réunion using 532 nm (Hoareau et al., 2012). The final design has been decided during a dedicated campaign that validated the instruments setup at the new Maïdo observatory (Keckhut et al., 2015). A detailed description of the actual system, the calibration method and the uncertainty budget can be found in Vérèmes et al. (2019) but the main characteristics are presented hereafter. The transmitter is based on two high energy pulse (375 mJ per pulse and a duration of 9 ns) synchronized Quanta Ray Nd:YAG lasers operating at 355 nm with a repetition

rate of 30 Hz. The receiver is a Newtonian telescope with a primary mirror of 1.2 m-diameter. The geometry of the transmitter and the receiver is co-axial to facilitate alignment and avoid parallax effects, thus extending the measurements to only a few meters above the ground. Because fluorescence in optical fibers can cause biases (Sherlock et al., 1999), no optical fiber is employed in Lidar1200. The Raman and the Rayleigh signals are separated by dichroic beam splitters and interference filters located directly after the telescope.

The water vapor profiles are calibrated using Global Navigation Satellite System Integrated Water Vapor (GNSS IWV) (Vérèmes et al., 2019). The calibration coefficient is systematically estimated every night and updated each time an instrumental change occurs, change typically identified by checking the results of the daily lamp measurements and the logbook overview. The "nightly coefficient" corresponds to the average for a night of measurement of the 5-min ratios between the GNSS IWV and the uncalibrated lidar IWV data. The system is able to measure water vapor up to 22 km a.s.l. by integrating

several nights of measurement whereas the maximum altitude for average profile of 240 min is around 15 km a.s.l. for a total uncertainty lower than 30 % (Vérèmes et al., 2019). The measurement schedule of Lidar1200 implies 2 measurement nights



per week in routine mode, to which we can add the campaigns measurements. The time slot of routine operation is 19:00 to 01:00 local time, depending on the meteorological conditions. The corresponding dataset is archived at the NDACC data handling center. For this study, only profiles with a random uncertainty lower than 15 ppmv at each vertical level between 16 and 20 km a.s.l. were used.

## 2.4 CALIOP Level 3 Stratospheric Aerosol Profile

Although TMWAL measures the atmospheric backscatter at 355 nm with three different receivers, their dynamic ranges are not matched in a way that allows accurate retrieval of atmospheric backscatter profiles in the UTLS. The 355 nm channel coupled to the largest receiver is optimized for temperature retrievals and gated below 22 km a.s.l., while the other two channels are small receivers tailored for measurements in the lower troposphere. Similarly, there is no NDACC-archived aerosol products from the other two lidars (RALMO and Lidar1200). For this reason, and in order to use a common criterion to evaluate the aerosol influence in the three systems used in this study, we decided to use the new CALIOP Level 3 (L3) stratospheric aerosol profile product (Kar et al., 2019), released in August 2018, to serve as correlative aerosol measurements at the three NDACC water vapor lidar sites. This spaceborne lidar product reports monthly mean profiles of aerosol extinction, particulate backscatter, attenuated scattering ratio (SR), and stratospheric aerosol optical depth on a spatial grid of 5° in latitude, 20° in longitude, and 900 m in altitude. As part of this dataset, two different aerosol products are reported. One is labeled as "background" and the other is labeled "all aerosols". While the first corresponds to profiles retrieved after removing clouds, aerosols, and polar stratospheric clouds (PSCs), the second only screens out clouds and PSCs. For this study, we use the all aerosols data product from dataset version 1.0 before July 2020 and version 1.01 since then.

## 2.5 Calibration by radiosounding

At TMF, Vaisala RS92 radiosondes were used systematically from the beginning of the lidar operations until 2013. The transition towards Vaisala RS41 radiosondes occurred between 2014 and 2018, during which either RS92, or RS41, or both radiosonde types were launched on a given night (Dirksen et al., 2020). Vaisala RS41 radiosondes have been exclusively used since 2018. The RS92 radiosonde profiles are corrected for time lag and dry bias following Miloshevich et al. (2009), which minimizes the impact of the radiosonde type change on the long-term time series.

At Payerne, the radiosounding are launched twice daily, at 11:00 UTC and 23:00 UTC, since almost 70 years. The calibration of the RALMO WVMR profiles has used the SRS-C34 sondes profile since 2008 until the beginning of 2017, transitioning first to the SRS-C50 model on February 2017, and finally to the Vaisala RS41 radiosondes on march 2018. In the case of the SRS-C34 sondes, the humidity profile could be used trustworthy within the instrument accuracy up to 10-12 km a.s.l. After switching to the SRS-C50 model this range could be extended slightly higher, although both radiosonde's and RALMO's accuracy in the UTLS remained limited. The calibration of the WVMR measured by RALMO is automatic and occurs every day in clear sky when the sun is at 19 degrees elevation angle. The automatic calibration uses as initial point a radiosounding calibration and adjusts this value recursively in time accordingly to the temporal drift of the ratio of the nitrogen to the water



vapor signal using the solar background. The reference calibration timeseries for the WVMR is the series published by Hicks-Jalali and colleagues (Hicks-Jalali et al., 2020) updated to 2021.

In the case of the Lidar1200 system in Reunion Island, there are no radiosondes regularly launched from the Maïdo Observatory. The closest operational radiosonde station is at the Saint-Denis airport, but the Meteomodem M10 sonde used there does not provide reliable humidity measurements in the UTLS. The UTLS part has been validated using

Cryogenic Frostpoint Hygrometer sondes (Vérèmes et al., 2019). However, the calibration was also based on the comparison of the with collocated global navigation satellite system integrated water vapor measurements.

## 3 Assessing the impact of fluorescence on the water vapor lidar retrievals

In order to evaluate the magnitude of possible biases affecting the TMWAL measurements in the UTLS, and investigate its temporal variability, we compared the TMWAL water vapor profiles with all available co-located radiosonde profiles since 2009. A total of 482 lidar profiles, 236 RS92, and 246 RS41 radiosonde launches were included in the comparison. The coinciding radiosondes and lidar profiles were averaged to a common grid with a vertical resolution of 1 km and a temporal resolution of one month. An overview of the lidar and radiosonde measurements between 10 and 20 km a.s.l. is presented in Fig. 1, together with the relative difference and the zonally averaged CALIOP stratospheric aerosol scattering ratio product.



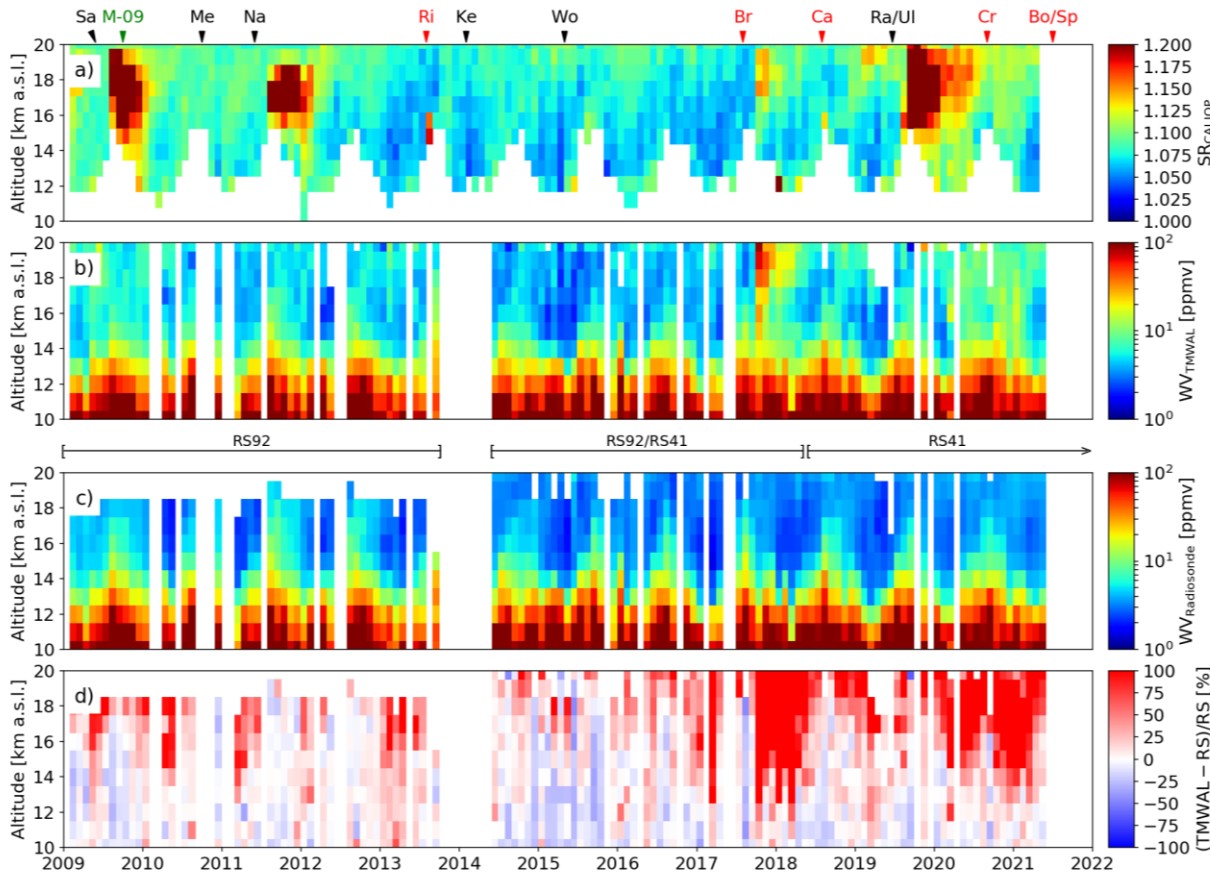

**Figure 1. (a)** Monthly and zonally averaged time series of aerosol scattering ratio derived from CALIOP between 2009 and end of 2021 at 30-35° N (Table Mountain Facility). Major eruptions (black), PyroCb events (red), and the MOHAVE-2009 field-campaign are indicated at the top of the plot (see Table 1). **(b)** Monthly-averaged TMWAL water vapor mixing ratio measurements during days where radiosonde launches were available. **(c)** Monthly-averaged radiosonde water vapor mixing ratio. The periods where the RS92, RS92 and RS41, and RS41 radiosonde were used are indicated on top of the plot. **(d)** Relative difference between TMWAL and the radiosondes.

**Table 1.** Most significative volcanic eruptions (Volcanic Explosivity Index (VEI) > 4) and PyroCb events affecting measurements over TMF, Payerne and Reunion Island, for the period comprehended between January 2009 and November 2022. Information regarding the location of the erupting volcano and their VEI was retrieved from the Global Volcanism Program (https://volcano.si.edu/, last access: 22 November 2021). In the case of the PyroCbs, available references are included in the last column.

| Volcano / PyroCb | Event date | Location | Reference |
|---|---|---|---|
| Sarychev (Sa) | Jun 2009 | Kuril Islands (48.1° N) | |
| Merapi (Me) | Oct 2010 | Indonesia (7.5° S) | |
| Nabro (Na) | Jun 2011 | Eritrea (13.4° N) | |
| Rim Fire (Ri) | Aug 2013 | California (37° N) | Peterson et al. (2015) |
| Kelud (Ke) | Feb 2014 | Indonesia (7.9° S) | |
| Calbuco (Ca) | April 2015 | Chile (41° S) | |





| Wolf (Wo) | May 2015 | Galápagos Islands (0.0° N) | |
| British Columbia (Br) | Aug 2017 | British Columbia (54° N) | Peterson et al. (2018) |
| Ambae (Am) | Sep 2017 | Vanuatu (15° S) | |
| Carr Fire (Ca) | Aug 2018 | California (41° N) | Lareau et al. (2018) |
| Ulawun (Ul) | Jun 2019 | Papua New Guinea (5.0° S) | |
| Raikoke (Ra) | Jun 2019 | Sea of Okhotsk (48.3° N) | |
| Australian Fires (Au) | Dec 2019 | Australia (34° S) | Khaykin et al. (2020) |
| Creek Fire (Cr) | Sep 2020 | California (37° N) | |
| Bootleg Fire (Bo) | Jul 2021 | Oregon (43° N) | |
| Sparks Lake Fire (Sp) | Jul 2021 | British Columbia (54° N) | |

As mentioned in Sec. 1, there are many different systematic error sources affecting Raman lidars measurements, and each of
them will have a specific way to affect the retrievals, making them generally discernible from each other. In the case of aerosol
extinction and insufficient blocking of Mie scattering, we would expect the impact on the measurements to be proportional to
the intensity of the aerosol backscatter properties, regardless of the aerosol type (i.e., volcanic plumes, smoke plumes, dust,
etc.). In case of fluorescence induced within the instrument optical components, we would expect a time-independent
contamination, whether aerosols are present or not. On the other hand, in the case of fluorescence induced by the aerosols, we
would expect a signature only in the presence of fluorescing aerosols.

When looking at the TMWAL and radiosonde time series, as well as the relative difference between the two presented in Fig.
1d, the most prominent disagreement appears towards the end of 2017 above 12 km a.s.l.. This appears simultaneously with a
strong increase in scattering ratio measured by CALIOP at a similar altitude range (Fig. 1a). This could suggest in principle
insufficient blocking of the Mie scattering, but three other strong aerosol events observed by CALIOP around mid-2009, mid-
2011, and the end of 2019 had a much larger scattering ratio, and yet the effect on the TMWAL water vapor profiles was
negligible. In particular, Leblanc et al. (2011) shows that measurements conducted during MOHAVE-2009 were simply
unaffected by the aerosol plume from the Sarychev eruption. This suggests that the bias observed in TMWAL was associated
with aerosol fluorescence rather than insufficient blocking of the elastic scattering. The three plumes, characterized by the
largest scattering ratio and negligible impact on the TMWAL water vapor series, correspond to the eruption of the Sarychev,
Nabro and Ulawun/Raikoke volcanos (Khaykin et al., 2017). The plume observed at the end of 2017 corresponds to smoke
aerosol particles injected into the lower stratosphere from a very notorious series of events, specifically five near-simultaneous
intense PyroCbs (pyrocumulonimbus) occurring in western North America on 12 August 2017 (Peterson et al., 2018). At the
end of 2018 and 2019 this wet bias decreases significantly (observable down to 16 km a.s.l.), and then increases again in 2020
and 2021 as a result of further smoke injection associated with large PyroCbs registered in western United States and Canada
(NASA Earth Observatory, 2021). The present results are consistent with a case study previously reported by Immler et al.



(2005), during which a Raman lidar located in Lindenberg (Germany) showed a water vapor wet bias inside a passing smoke plume that originated from wildfires in Portugal.

In order to further support our findings on the origin of the observed biases, we conducted a similar analysis on the RALMO (Fig. 2) and Lidar1200 (Fig. 3) lidar datasets. Because of the zonal symmetry and widespread nature of the volcanic and smoke

plumes considered here in the northern hemisphere mid-latitudes, we expect the RALMO dataset to be influenced in a similar way as TMWAL's dataset. Although the uncertainty related to RALMO humidity measurements increases rapidly above 12-14 km a.s.l., we can clearly see a positive bias in the lidar-measured water vapor towards the end of 2017 and the last months of 2021 (Fig. 2b) with respect to the co-located radiosoundings (Fig. 2c). Just like for TMWAL, the 2017 wet bias is coincident with an increase of scattering ratio retrieved from CALIOP (Fig. 2a), which was traced back to the intense wildfires in British

Columbia in summer 2017 (Peterson et al., 2018). The apparent RALMO water vapor increase is not observed by the co-located Payerne radiosondes, which suggests again a contamination of the water vapor lidar signal by fluorescing smoke. As in TMWAL, the RALMO water vapor dataset does not show any wet bias after the Raikoke/Ulawun eruption despite the large scattering ratio observed by CALIOP, confirming that insufficient blocking of Mie scattering by RALMO can be ruled out.

As for the Lidar1200 lidar at Reunion Island (southern hemisphere tropics), Fig. 3b exhibit the same behavior as for the other

two lidars. There is no apparent impact of the Calbuco volcanic aerosol plume observed by CALIOP over Reunion Island during the 2015-2016 period. On the other hand, a strong apparent water vapor increase is observed by Lidar1200 during the second half of 2020 between 18 and 22 km a.s.l., increase coinciding with the presence of the smoke plume produced by the January 2020 Australian fires. Although no radiosonde measurement is available at Reunion Island, the water vapor values observed by the lidar (>40 ppmv) are unrealistically high, even considering the humidification of the UTLS caused by the

smoke plume injection (Khaykin et al., 2020).

Based on the results presented in this section, we can conclude that fluorescence in smoke plumes transported in the UTLS induces substantial wet biases in the water vapor UV Raman lidar retrievals at different locations in the northern and southern hemispheres. The wet biases remain as long as the fluorescing aerosols remain in the UTLS after their injection across the tropopause. It is important to point out that, while large PyroCb events induce signatures on the water vapor lidar profiles that

are easy to identify, smaller fire events produce smaller signatures that are more difficult to identify. Besides their reduced intensity, these events are more difficult to characterize as the associated smoke plumes may have pathways that differ significantly from the typical deep convection induced by large fires. Smaller smoke injection events can potentially induce wet biases up to a few percent, very difficult to distinguish from other uncertainty sources and biases, but large enough to affect the accuracy of trend detection. In an effort to minimize this impact, a correction technique relying on the addition of a

dedicated "fluorescence channel" was developed. Details of this correction technique are presented in the following section.



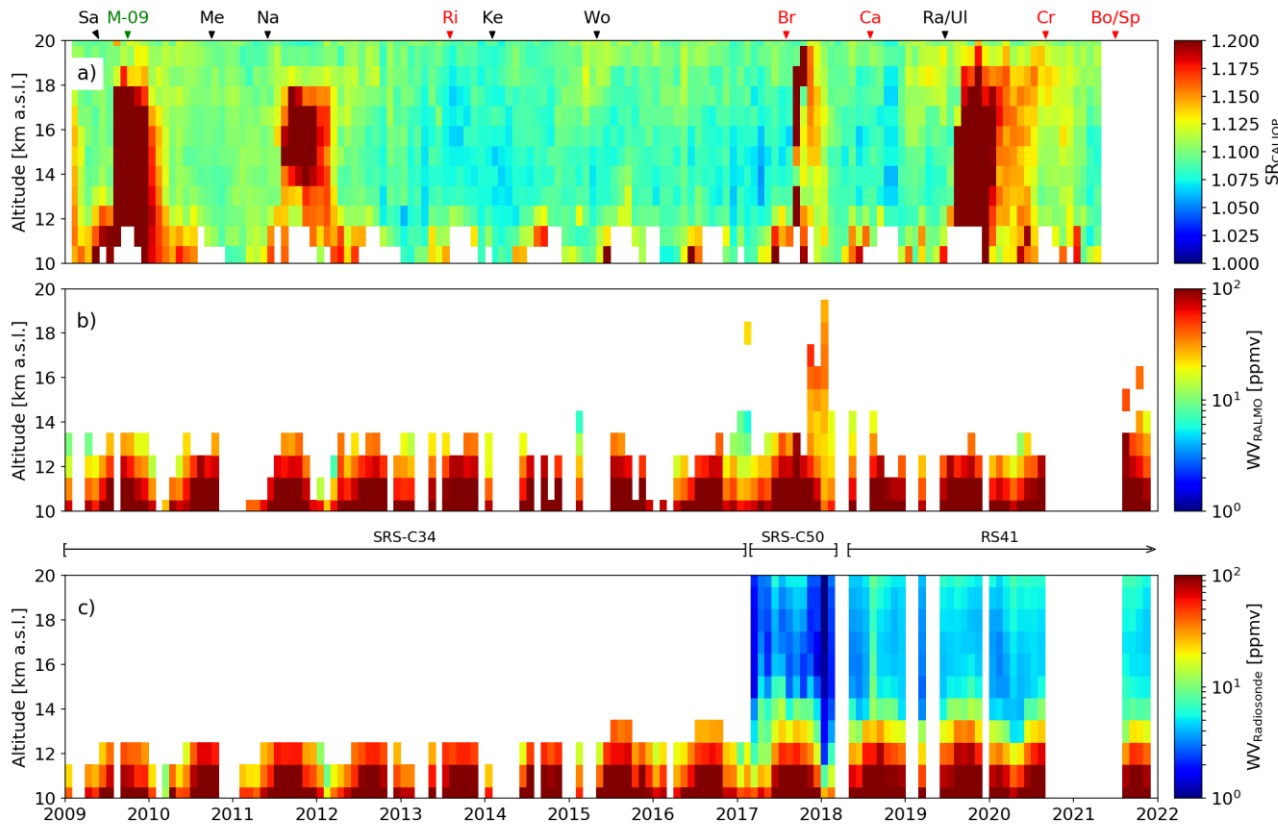

**Figure 2. (a)** Monthly and zonally averaged time series of aerosol scattering ratio derived from CALIOP between 2009 and end of 2021 at 45-50° N (Payerne). Major eruptions (black), PyroCb events (red), and the MOHAVE-2009 field campaign are indicated at the top of the plot (see Table 1). **(b)** Monthly-averaged RALMO water vapor mixing ratio measurements during days where radiosonde launches were available. **(c)** Monthly-averaged radiosonde water vapor mixing ratio. The periods where the SRS-C34, SRS-C50, and RS41 radiosondes were used are indicated on top of the plot.



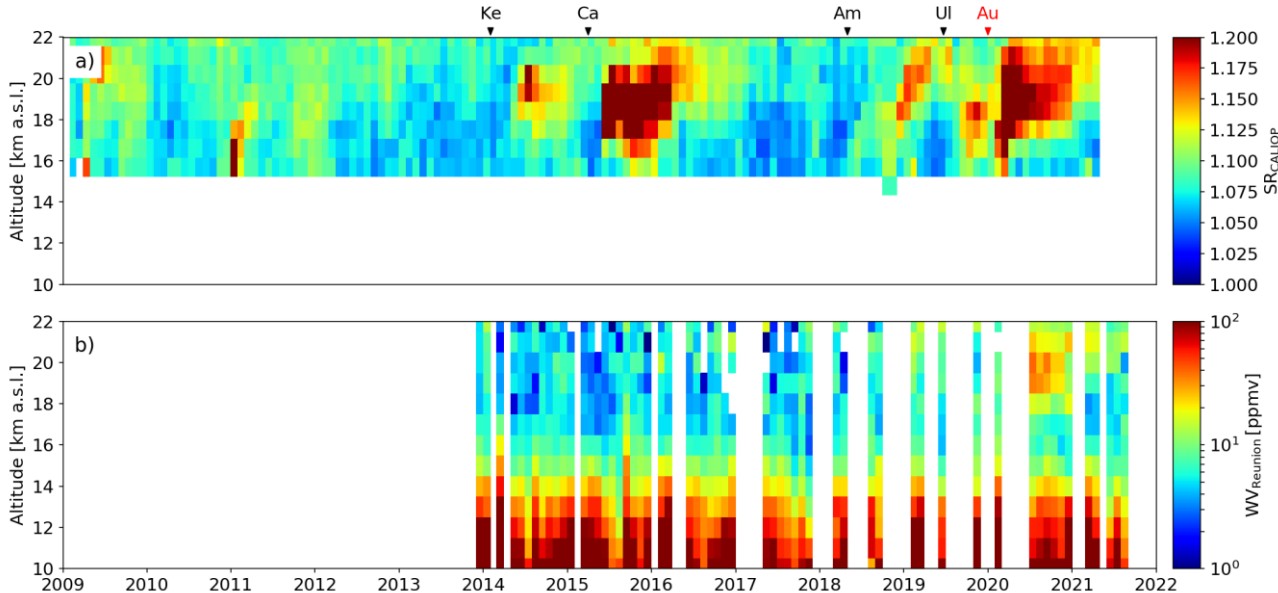

**Figure 3. (a)** Monthly and zonally averaged time series of aerosol scattering ratio derived from CALIOP between 2009 and end of 2021 at 25-20° S (Reunion Island). Major eruptions (black), PyroCb events (red) are indicated at the top of the plot (see Table 1). **(b)** Monthly-averaged Lidar1200 water vapor mixing ratio.

## 4 Proposed correction

### 4.1 Method

In the typical UV water vapor Raman lidar technique, the emission wavelength is at 355 nm and the water vapor signal is collected near 407.5 nm (Fig. 4), which corresponds to the Stokes Q-branch of the OH-stretching band (~3654 cm$^{-1}$). In a dry clean atmosphere, most of the water vapor Raman-backscattered intensity is confined within +/-0.5 nm of the strongest line, and after proper rejection of the much more intense elastic signal, the water vapor Raman signal is the only signal left between 406 and 410 nm. Yet, when the transmitted beam (355 nm) probes a layer of biogenic aerosol (smoke plume), an additional signal is produced, namely fluorescence, a broadband signal which spectrum starts near the emission wavelength and extends well beyond the region of the water vapor Raman lines. In this case, the signal collected in the water vapor channel (407.5 nm) becomes the sum of the water vapor Raman backscatter signal and the fluorescence signal. The exact spectral shape of the fluorescence spectrum depends on the aerosol composition, and therefore is typically unknown, with the exception that it is broadband, i.e., the spectrum varies slowly with wavelength and therefore can be assumed nearly-flat in the region 406-412 nm. With these considerations in mind, we can then design a new, dedicated "fluorescence channel", which purpose is to estimate the intensity of fluorescence, and subtract it from the contaminated water vapor signal.



In an ideal scenario, we would like to have the water vapor and fluorescence channels set up so that they would each collect signal from their respective spectrum only. This is not possible because of the broadband nature of the fluorescence spectrum, so the idea is to pick a center wavelength for the fluorescence channel that is very close to the water vapor Raman lines, yet keeping the water vapor contribution as small as possible. We therefore selected a fluorescence filter with a center wavelength

of 410.3 nm, a bandwidth of 1.1 nm FWHM (Full Width at Half Maximum), and a peak transmission of about 90%. The center wavelength is less than 3 nm away from the center wavelength of the water vapor Raman channel, which allows us to assume a nearly identical magnitude of the fluorescence signal entering the water vapor and the fluorescence channels. This assumption seems to be reasonable based on several studies conducted on the fluorescence characteristics of biomass burning aerosols, that show a broad slow-varying emission spectrum when biomass burning aerosols are excited at 355 nm (e.g. Pan et al., 2007;

Fu et al., 2015; Tang et al., 2020). At the same time, the intensity of the water vapor Raman scattering in the new fluorescence channel is attenuated by about an order of magnitude when compared to the water vapor Raman channel (Fig. 4).

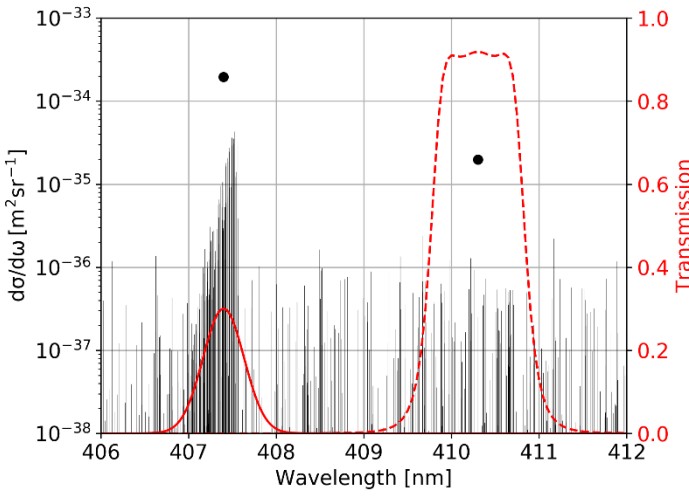

c

**Figure 4.** Water vapor Raman cross-section spectrum (black) is shown together with the transmission of the water vapor channel filter (red)
and the new fluorescence monitoring channel filter (dashed, red). The total water vapor Raman cross-section for these two channels as the result of integrating the product of the Raman cross-section spectrum with the filter transmission of each channel between 406 nm and 412 nm is also shown (dots, black).

The addition of the fluorescence channel had to be done with a minimum disturbance to the existing receiver, and also at the lowest cost possible. Adding a dichroic beam splitter to keep the 407.5 nm on one end, and add a new 410.3 nm channel on

the other end was a possibility. However, the small spectral difference (3 nm) between the two channels would have made this option challenging, and the need for an additional photomultiplier tube (PMT) and its downstream electronics would have made this option less affordable. To reduce both cost and complexity, it was therefore decided to use a filter wheel instead. The new filter wheel is mounted in place of the original 407.4 nm interference filter, and now includes both the original filter at 407.4 nm and an additional 410.3 nm filter for the fluorescence data. The stability of the system is maximized since the

same 407.4 nm filter, same PMT and same electronics are used for the water vapor channel. The new fluorescence channel actually uses the same hardware as the water vapor channel, except for the filter on the filter wheel. The data is acquired alternatively for each filter as the wheel rotates through a pre-defined duty cycle. To minimize the effect of the atmospheric variability between the acquisition of the water vapor data and the fluorescence data, an interleaved acquisition scheme was implemented. Instead of the regular 2-hour experiments, the results shown in Sec. 4.2 are the result of 4-hour acquisitions,

where the filter wheel is changed between the two filters every 30 minutes as part of our automated measurement routine.

The quantification of the fluorescence signal with respect to the water vapor signal is done not only by measuring the absolute signals in each channel, but also by inter-calibrating the two channels. The number of photons measured at a given range gate $R$ on the water vapor Raman channel ($N^{407}$) can be written as

$$N^{407}(\text{R}) = N_{WV}^{407}(R) + N_F^{407}(R) + N_{BKG}^{407} \text{ ,} \tag{1}$$

$N_{WV}^{407}$ represents the contribution from water vapor Raman scattering, $N_F^{407}$ represents fluorescence, and $N_{BKG}^{407}$ represents the background contribution (dominated by moonlight and sky background rather than PMT dark counts).

Similarly, the number of photons measured on the new fluorescence channel $N^{410}$ at the same range gate can be written in

terms of the same components as

$$N^{410}(\text{R}) = \text{k}_L^{407 \rightarrow 410} \cdot (\text{k}_{WV}^{407 \rightarrow 410} \cdot N_{WV}^{407}(\text{R}) + \text{k}_F^{407 \rightarrow 410} \cdot N_F^{407}(\text{R}) + \text{k}_B^{407 \rightarrow 410} \cdot N_{BKG}^{407}) \tag{2}$$

where $\text{k}_{WV}^{407 \rightarrow 410}$, $\text{k}_F^{407 \rightarrow 410}$ and $\text{k}_B^{407 \rightarrow 410}$ are proportionality constants given by the spectral characteristics of the water vapor

Raman, fluorescence and background signal, respectively. As mentioned before, we assume that the fluorescence spectra of the aerosols affecting the water vapor measurements can be considered to be constant over the 3 nm that separate the water vapor and the fluorescence monitoring channels, making $\text{k}_F^{407 \rightarrow 410} = 1$. On the other hand, $\text{k}_L^{407 \rightarrow 410}$ is the lidar spectral response ratio and corresponds to the change in the lidar efficiency between these two wavelengths due to differences in filter transmission, filter width, PMT efficiency, etc. If we pick a range $R_{BKG}$ where no water vapor or fluorescence contribution is

expected ($N_{WV}^{407}(R_{BKG}) = 0$ and $N_F^{407}(R_{BKG}) = 0$), (1) and (2) can be written as:

$$N^{407}(R_{BKG}) = N_{BKG}^{407} \tag{3}$$

$$N^{410}(R_{BKG}) = \text{k}_L^{407 \rightarrow 410} \cdot \text{k}_B^{407 \rightarrow 410} \cdot N_{BKG}^{407} \tag{4}$$


Since the wavelength difference between the $N^{407}$ and $N^{410}$ channels are small, the background radiation at these two wavelengths are expected to be very close, and thus $\text{k}_B^{407 \rightarrow 410} = 1$. In order to check this assumption, we retrieved the top of





the atmosphere (TOA) moonlight spectra based on the algorithm provided by Miller and Turner (2009). The 410/407 nm spectral ratio at the TOA is 1.03, and mostly independent from the moon phase. If we then assume that the atmospheric

extinction is dominated by Rayleigh scattering around 400 nm (Cramer et al., 2013) and use a direct viewing geometry, we

obtain $k_B^{407 \to 410} = 1.03 \cdot \left(\frac{407 \text{ nm}}{410 \text{ nm}}\right)^4 = 1$. While the direct moon viewing does not correspond to a real observation geometry,

we expect the deviations from this assumption to be small based on other scattered moonlight model results (Jones et al., 2013).

Based on this assumption, we can derive our lidar spectral response ratio as


$$k_L^{407 \to 410} = \frac{N^{410}(R_{BKG})}{N^{407}(R_{BKG})} \tag{5}$$

Because $k_L^{407 \to 410}$ is related to the characteristics of the lidar components, we expect it to remain mostly constant over time, as long as there is no change in the lidar setup. Also, as mentioned before, this calculation assumes that the background signals

are dominated by moonlight sky background and that the contribution of the PMT dark counts and other light sources with unknown spectral characteristics is negligible. In order to verify this assumption, we analyzed the stability of the ratio defined in Eq. 5 as a function of the background levels.



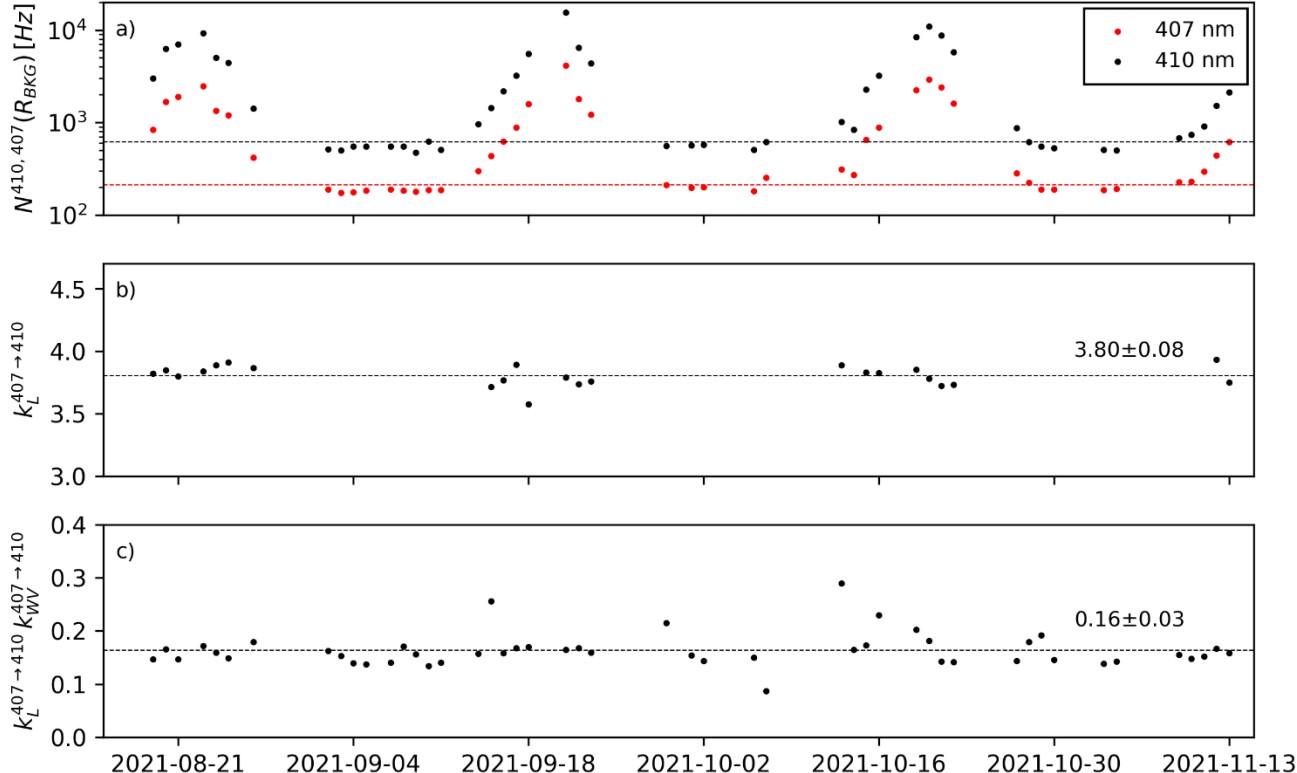

**Figure 5.** (a) Time series of background levels for the 407 nm (dots, red) and 410 nm (dots, black) channels. The average of the background levels for days where other unknown light sources dominate (low moonlight contribution) is also shown (dashed). (b) Time series of the 410/407 nm channel background ratio (lidar spectral response ratio) calculated based on days where moonlight dominates and after subtracting the contribution from unknown light sources. The average of the lidar spectral response ratio is also shown (dashed, black). (c) Time series of the ratio between the 410/407 nm channels for altitudes between 6-8 km a.s.l., where the fluorescence contribution can be typically considered negligible (Eq. 6). The average is also shown (dashed, black).

The background levels presented in Fig. 5a show a clear temporal pattern caused by the moon phases, with a maximum

background occurring during the full moon phase. As the moon illumination drops towards new moon, we can see that the

background levels stabilize to relatively constant values, while maintaining a substantial difference between the 407 nm and

410 nm channels. Since these two channels share the same PMT and optical path (with the exception of a changing filter), we

can conclude that these background levels are not related to PMT dark counts but rather to a different background source (i.e.

light pollution from Los Angeles basin and other nearby sources). Since the calculation of $k_L^{407\rightarrow410}$ assumes the knowledge

of the background spectral characteristics, we first estimated the contribution of the light pollution as the average of the

background for days where the moonlight contribution is negligible (Fig. 5a). This average was then subtracted from the

background levels of moon dominated days in order to keep only the moonlight contribution, that has a known spectrum. The

ratio of these two channels after correction (Fig. 5b), exhibit a fairly constant value, with an average of 3.8±0.08.




Next, the constant term $k_{WV}^{407\to410}$ can be determined by calculating the ratio of the 407 nm and 410 nm signals (after background subtraction) for a range R at which the fluorescence contribution is negligible, or else using laboratory measurements of the water vapor Raman spectrum. From Eqs. 1 and 2, after background subtraction (denoted by *), and assuming $N_F^{407}(R) \ll N_{WV}^{407}(R)$


$$k_L^{407\to410} \cdot k_{WV}^{407\to410} = \frac{N^{410*}(R)}{N^{407*}(R)} \tag{6}$$

Figure 5c shows the result of applying Eq. 6 to each measurement period between 9-11 km a.s.l., where $N_F^{407}(R) \ll N_{WV}^{407}(R)$. This ratio shows a relatively stable behavior and a mean value of 0.16±0.03, which translates into $k_{WV}^{407\to410} = 0.042$. This

result is relatively close to the one obtained by the other proposed method (laboratory measurements of the water vapor Raman cross-section) and shown in Fig. 4 as the two black dots with a ratio of about 0.1. The difference between these two approaches can be attributed to several different factors not accounted for in the calculation, like departure of the filter transfer functions from the values used for the calculation, changes in the PMT response function between the two wavelengths, etc.

After background subtraction we have the following equation system, where we can obtain the fluorescence-corrected water vapor signal needed to conduct the standard Raman lidar water vapor retrieval:

$$N^{407*}(R) = N_{WV}^{407}(R) + N_F^{407}(R) \tag{7}$$

$$N^{410*}(R) = k_L^{407\to410} \cdot k_{WV}^{407\to410} \cdot N_{WV}^{407}(R) + k_L^{407\to410} \cdot N_F^{407}(R) \tag{8}$$

From Eq. 7

$$N_{WV}^{407}(R) = N^{407*}(R) - N_F^{407}(R) \tag{9}$$


While from Eq. 8

$$N_F^{407}(R) = \frac{1}{k_L^{407\to410}} N^{410*}(R) - k_{WV}^{407\to410} \cdot N_{WV}^{407}(R) \tag{10}$$

Replacing Eq. 10 in Eq. 9, we obtain an expression of the "true" water vapor contribution (i.e., the corrected water vapor profile), $N_{WV}^{407}(R)$:





$$N_{WV}^{407}(R) = \frac{N^{407*}(R)}{(1 - k_{WV}^{407 \to 410})} - \frac{N^{410*}(R)}{k_L^{407 \to 410}(1 - k_{WV}^{407 \to 410})} \tag{11}$$

**4.2 Correction results and uncertainty**

To evaluate the performance of the fluorescence correction method described in the previous section, we conducted at JPL-
TMF a series of 22 experiments between 27 August and 4 November 2021, where the uncorrected and corrected TMWAL
profiles were compared with co-located RS41 radiosonde launches. The results of these tests are summarized in Fig. 6.



**Figure 6.** (a) The average of the 22 profiles measured between 27 August and 4 November 2021 by TMWAL before the correction method
was applied (solid, blue), after the correction method was applied (solid, orange) are shown together with the average of the co-located RS41
radiosonde profiles (solid, green). The retrieved TMWAL profiles are calibrated using the RS41 measurements between 9 and 14 km a.s.l.
(red stripe). (b) Average of the relative difference between the uncorrected (solid, blue) and corrected (solid, orange) TMWAL profiles when
compared with the co-located RS41 launches. (c) Average random uncertainty (1-sigma) for the uncorrected (solid, blue) and corrected
(solid, orange) TMWAL profiles.

Between 14 km and 20 km a.s.l., the average of the uncorrected TMWAL profiles (blue curve) shows a clear wet bias ($> 50\%$)
with respect to the RS41 mean profile, reaching a maximum of 75% between 17 km and 18 km a.s.l.. This is similar to the
biases observed before the correction method was implemented towards the end of August 2021 (Fig. 1d). The average of the



corrected profiles (orange curve) shows a much better agreement with the RS41 mean profile, within 10% for most of the altitudes, and within 5% overall. Above 18 km a.s.l., the difference with the RS41 increases to about 25%. Above 19.3 km a.s.l., the corrected profile gets drier than the RS41 profile, but the low signal-to-noise ratio at these altitudes makes the

difference clearly statistically not significant. When only profiles measured during low background conditions (new moon) are used, this deviation at the top of the profile is substantially reduced (not shown).

The main factors controlling the performance of the proposed correction method is the accuracy in the determination of the $k_L^{407\rightarrow410}$ and $k_F^{407\rightarrow410}$ constants. These two constants have a similar impact on the final result, but $k_F^{407\rightarrow410}$ is substantially more difficult to quantify and much more variable as it depends on the composition of the interfering aerosols.

The next consideration is the expected increase of the corrected water vapor random uncertainty. The Raman lidar technique uses the ratio of signals backscattered by water vapor and nitrogen. In the UTLS, the random noise is dominated by the water vapor channel detection noise since the nitrogen signal is several orders of magnitude larger than the water vapor signal. With the assumption that the water vapor and fluorescence components are independent, the variance of the corrected water vapor signal is the sum of the original water vapor signal variance and the fluorescence signal variance divided by the lidar spectral

response ratio. At altitudes above 16 km, where the fluorescence signal is of the same order of magnitude as the water vapor signal, we expect a substantial increase in the random uncertainty of the corrected retrieval. In order to verify this, we conducted a Monte Carlo (MC) simulation. The MC uncertainty simulation presented in Fig. 6c is the result of 100 realizations, where each realization models all contributing uncertainties (calibrations and detection noise) as normally distributed. While the detection noise of each individual detected shot follows a Poisson distribution, the average detection noise of each experiment

(realization) can be modeled as normally distributed following the central limit theorem. Figure 6c shows that the random uncertainty of the corrected TMWAL profiles is about 50% larger than that of the uncorrected profiles over the altitude range where the correction is most important. The increase of the random uncertainty affects the ability of the system to detects potential trends (Whiteman et al., 2011), and additional measurements would typically be required to mitigate this degradation. Yet, the systematic uncertainty introduced by the fluorescence correction (determination of $k_L^{407\rightarrow410}$ and $k_F^{407\rightarrow410}$ constants)

remains the most critical component for the accurate estimation of long-term trends.

## 5 Discussion and conclusions

Almost two decades after being added to the suite of NDACC instrumentation, the Raman water vapor lidar technique is facing a new, unforeseen challenge with the recent increase of lower stratospheric aerosol loading. In the northern hemisphere, sustained droughts in North America and Siberia were responsible for large and more frequent wildfires since 2017. Many of

these fires led to substantial PyroCb events, resulting in the injection of unprecedented quantities of smoke in the northern hemisphere lower stratosphere. Large injections of smoke also occurred in the southern hemisphere lower stratosphere following the Australian wildfires in the austral summer 2019/2020. When smoke layers in the UTLS are sounded by water vapor Raman lidars transmitting in the UV (355 nm), they fluoresce and can cause significant contamination of the weak water



vapor lidar signals collected at these altitudes. Such contamination was identified in the present study using the long-term
water vapor datasets of three ground-based UV water vapor Raman lidars contributing to NDACC, together with the aerosol
scattering ratio profiles from the spaceborne CALIOP lidar. Between 2009 and 2017, no large wildfire event was identified,
and only volcanic eruptions have caused observable injection of aerosols in the stratosphere. Over this period, the TMWAL
lidar was able to provide excellent quality water vapor profiles, typically up to 20 km. But since 2017, the impact of the
increased wildfire activity on the TMWAL, RALMO and Lidar1200 water vapor data was clearly identified in the UTLS, with
a high bias of up to 75% at 17 km for TMWAL. As a result of this contamination, a large fraction of the TMWAL water
profiles had to be cut-off below 15 km before they were archived at NDACC.

As of today, it is difficult to predict if this increased fire activity is temporary, or if it will become the norm for the next decades.
In any case, it has raised the important question of the usefulness of UV water vapor Raman lidar measurements for the
detection of long-term trends in the UTLS. Our present study shows that a low-cost upgrade of the lidar receiver allows for a
fluorescence correction. Our results are encouraging, and allowed us to basically remove the fluorescence contribution from
the contaminated lidar signals. The improvement results in a drastic reduction of the water vapor wet bias (up to 75% between
17-18 km a.s.l.), leading to a difference of only 5% with our co-located radiosonde measurements. However, this correction
was possible at the cost of increasing the total uncertainty, making the accurate detection of trends in the UTLS questionable.
Using the TMWAL data, it is shown that the relative impact of the fluorescence increases abruptly at around 15 km (Fig. 6b).
It is therefore reasonable to think that the detection of water vapor long-term trends by UV Raman lidars will only be slightly
impacted in the upper troposphere where the water vapor mixing ratio is larger than 20 ppmv, but will likely be very challenging
in the lower stratosphere without implementing very accurate fluorescence correction techniques.

The next question therefore is to figure out if a different lidar configuration can lead to a more robust water vapor measurement
in the UTLS. Different alternatives have been evaluated. One option is to add a second fluorescence channel with a center
wavelength slightly lower than the existing Raman water vapor channel. By doing so, there is the potential to reduce the
uncertainty in the shape of the fluorescence spectrum in the region of the water vapor Raman lines. This better quantitative
estimation of fluorescence will reduce the correction uncertainty and therefore total uncertainty on the corrected water vapor.
Another option is to add a second water vapor channel using a filter of broader bandwidth. This option however would make
the retrieval more sensitive to the variability of $k_w$, as both channels will have a similar contribution of water vapor Raman
scattering. Another possible, but much more expensive option is the addition of a spectrometer. This type of instrumentation
allows, in principle, an accurate quantitative estimation of the fluorescence spectrum in the entire region 406-412 nm, and once
again would reduce the corrected water vapor uncertainty.

Unfortunately, none of the lidar receiver upgrades just proposed provides the certainty to drastically improve the water vapor
measurement in the UTLS. As an alternative to implementing a fluorescence correction, the modification of TMWAL to
operate at a different wavelength is now considered. Although not negligible, aerosol fluorescence in the UTLS is expected to
be much smaller when the aerosol layer is excited at 532 nm. Using the YAG second harmonic would increase the transmitted
power by a factor of two, but the wavelength dependence of the Raman and Rayleigh scattering, the generally lower



performance of PMTs in that spectral region, and the expected higher sky background noise would result in the end in a reduction of the signal-to-noise ratio by a factor of about two (Sherlock et al., 1999) compared to our UV (355 nm) transmitter.

This option has a significant potential, but its implementation at JPL-TMF is expected to be costly and time consuming. Meanwhile, several NDACC Raman lidars currently operate at 532 nm. Alternatively, we are also evaluating the possibility of using a XeCl excimer laser (308 nm) as our source (Klanner et al., 2021). While generally speaking larger impact of aerosol fluorescence is expected when operating at shorter wavelengths, this is not necessarily true for the type of components responsible for the observed fluorescence interference in the UTLS. Benzo[a]pyrene, one of the components typically found

in smoke and mentioned by Immler et al. (2005a) as the potential responsible for the observed fluorescence contamination, exhibit very little emission under 350 nm (Fernández-Sánchez et al., 2003). While such system would profit form stronger Raman backscatter and lower sky background when compared to the YAG second harmonic alternative, it might have a stronger interference with ozone and other type of atmospheric aerosols.

An increased scrutiny of their measurement behavior in the UTLS is being called for. Discussion within the NDACC

community was initiated, with the main objective to foster close collaboration with key NDACC water vapor lidar groups using a transmitter at 532 nm or 308 nm, and assess the sensitivity of these lidars to aerosol fluorescence in the UTLS. Once this assessment is complete and successful, the scientific community in general, and the NDACC community in particular, will need to make important decisions regarding their UTLS water vapor long-term measurement strategy for the upcoming decades.


**Data availability**

TMWAL data is publicly available at https://www-air.larc.nasa.gov/missions/ndacc/data.html?station=table.mountain.ca/hdf/lidar/ (last access 15 February 2022). RALMO data is publicly available at https://www-air.larc.nasa.gov/missions/ndacc/data.html?station=payerne/hdf/lidar/ (last access

14 March 2022). The 2013-2018 OPAR Lidar1200 dataset is publicly available at https://www-air.larc.nasa.gov/missions/ndacc/data.html?station=la.reunion.maido/hdf/lidar/ (last access 11 February 2022). The 2019-2021 dataset is available under open-access by request to valentin.duflot@univ-reunion.fr. CALIOP L3 data is available at https://earthdata.nasa.gov/ (last access 15 February 2022).

**Author contributions**

Fernando Chouza prepared most of the manuscript and conceptualized the proposed correction method. Thierry Leblanc is the principal investigator of TMWAL, and contributed to the data analysis and development of the correction method. Mark



Brewer and Patrick Wang contributed in the technical implementation and validation of the proposed correction method. Giovanni Martucci and Alexander Haefele provided the RALMO measurements and contributed to the discussion of the results. Hélène Vérèmes, Valentin Duflot, Guillaume Payen and Philippe Keckhut conducted the analysis of the Lidar1200 data and contributed to the discussion of the presented results.

**Competing interests**

The authors declare that they have no conflict of interest.

**Acknowledgments**

The research was carried out at the Jet Propulsion Laboratory, California Institute of Technology under a contract with the National Aeronautics and Space Administration (80NM0018D004). The authors also acknowledge the European Communities, the Région Réunion, CNRS, and Université de La Réunion for their support and contribution in the construction phase of the research infrastructure OPAR (Observatoire de Physique de l'Atmosphère de La Réunion). OPAR is presently funded by CNRS (INSU), Météo France, and Université de La Réunion, and managed by OSU-R (Observatoire des Sciences de l'Univers de La Réunion, UAR 3365). OPAR is supported by the french research infrastructure ACTRIS-FR (Aerosols, Clouds, and Trace gases Research InfraStructure - France).

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
