# Peer review of "The impact of aerosol fluorescence on water vapor long-term monitoring by Raman lidar and the evaluation of a potential correction method"

_Atmospheric Measurement Techniques, 2022_

## Referee Comment (RC2)

Referee: Dr. Jens Reichardt, MOL/RAO, DWD, Germany.

**Summary:** The authors report on the effect of aerosol fluorescence on water vapor measurements in the lower stratosphere with high-performance lidars of the global NDACC, and on an instrumental method for correcting it. First, long time series of monthly means of water-vapor measurements with lidar and radiosonde at three sites are correlated with satellite aerosol data to make the case that lidar-derived stratospheric humidity exhibits a significant moist bias if biomass-burning aerosol (BBA) is present. It is concluded that this systematic error is caused by BBA fluorescence. To correct for this adverse effect, an instrumental technique is developed that is based on an interference filter with center wavelength close to the dominant water-vapor Raman line which is alternatingly exchanged for the water-vapor interference filter for quasi-simultaneous fluorescence detection. First measurements after a careful characterization of the technical upgrade show a significant reduction of the wet bias in the lower stratosphere. However, it comes at a price, which is a non-negligible increase in measurement uncertainty. For this reason, alternative approaches to a fluorescence correction are discussed.

**General comments:** The study presents interesting and important results, is well written and deserves publication after only some minor corrections. Some additional thoughts on the matter: (1) The authors had to expend great effort to establish the working hypothesis that fluorescence by stratospheric BBA caused the wet bias. Monthly averages and satellite aerosol data seem to be sufficient, but how much easier would the study have been if some simultaneous lidar aerosol data had been available. All lidar described are extremely potent instruments, so it should be relatively easy to augment the lidar data set with aerosol parameters, if budgets permit. (2) The referee fully agrees with the assessment given concerning the shape of the BBA fluorescence spectrum and the potential use of a spectrometer (or at least two discrete detection channels on both sides of the main watervapor Raman line) to account for spectral gradients around 407 nm. Measurements with the spectrometric RAMSES lidar reveal just that in BBA filaments at the tropopause (see figure). At 11.5 km (red curve), the spectral backscatter coefficient at 407.5 nm is 7%-8% smaller than at 410.5 nm. So in this case the current correction scheme would have introduced a dry bias to the mixing-ratio results.

Fig.: RAMSES measurement on 29 Sep 2021. The upper three curves depict spectra at different altitudes in a BBA filament at the tropopause (~11 km). BSC – backscatter coefficient

**Specific comments:**

I. 248: Given the scarcity of stratospheric measurements a slightly more defensive wording would be warranted. Probably, the reason why there are data points at all is that the additional fluorescence signal pushes the statistical errors of apparent water vapor mixing ratio below the error threshold.

I. 253: Here the fluorescence is missing, hence, no (additional) signal in the water-vapor detection channel and thus the statistical errors remain high and above the error threshold. Maybe this should be mentioned.

Is. 293, 301: See 'general comments' above. If fluorescence increases with wavelength, the corrected water vapor mixing ratio will be too low.

Fig. 6: It is quite amazing that the aerosol effect is seen in the entire lower stratosphere up to almost 20 km. Did BBA get up that high in California? For instance, in northern Germany the maximum height of BBA detection reached ~16 km only once in 2020 and was always below ~13 km in 2021. A similar plot for a time period without stratospheric BBA contamination would probably be worthwhile. Further, mixing ratios in the upper troposphere over California are already so small that significant BBA fluorescence should interfere here with the humidity measurement as well. The reviewer observes this phenomenon in Germany quite often. Do you have such measurement cases? Finally, is the RS41 data product in the stratosphere actually good enough for this comparison?

**Technical corrections:**

I. 58: Aerosol typing is also discussed in: Reichardt, J., R. Leinweber, and A. Schwebe, 2018: Fluorescing aerosols and clouds: Investigations of co-existence. EPJ Web Conf., 176, 05010, https://doi.org/10.1051/epjconf/201817605010.

I. 171: ... product from...

- I. 189: trustworthily
- I. 200: ... of the with... (Something is missing here.)
- I. 330 ff.: Subscripts that are not variables must be in normal text style, not italic.
- I. 343: See 'general comments' above.

Fig. 5c), I. 393: Is the range 6-8 km or 9-11 km?

---

## Author Comment (AC1)

Dear reviewers

We would like to thank you for valuable comments on our manuscript. The original comments are in bold, followed by our answers and modifications introduced to the manuscript.

**Comments by Anonymous Referee #1**

**Fig.1. Green letters M-09 at the top of the figure probably mean MOHAVE-2009. This should be explained in the capture.**

We added '(green)' as a clarification to Fig. 1a caption:

"Major eruptions (black), PyroCb events (red), and the MOHAVE-2009 *(green)* field-campaign are indicated at the top of the plot (see Table 1)."

**Influence of fluorescence on the vapor measurements discussed also in: Reichardt, J.: Cloud and aerosol spectroscopy with Raman lidar, J. Atm. Ocean. Tech., 31, 1946-1963, 2014.**

We added a reference to that work as part of the following sentence:

"The first report of aerosol fluorescence contamination on a water vapor Raman lidar measurement was presented by Immler et al. (2005a) and further discussed by Immler et al. (2005b) and *Reichardt (2014)*."

**Fig.5c. From the title of left axis it is not clear, that this is the ratio of the channels**

Since Fig. 5c is the ratio of the channels after background subtraction we prefer to leave the notation as is. A reference to Eq. 6 explaining the notation is included in the figure caption.

**Fig.6c. I am a bit confused by the explanation of increase of uncertainty at high altitudes after correction. If we measure the sum of water vapor and fluorescence signal, does it mean that statistical uncertainty of vapor measurements becomes lower? I think uncertainty of vapor signal should stay the same…Correct me if I am wrong.**

If we simplify the problem (e.g. the instrumental response is the same for the WV and fluorescence channel, there is no background or cross-talk), the signal measured by the WV channel ($N_{407}$) is the sum of the water vapor Raman signal ($N_{WV}$) and the fluorescence signal $N_F$.

$$N_{407} = N_{WV} + N_F$$

While in the fluorescence monitoring channel ($N_{410}$)

$$N_{410} = N_F$$

To calculated the corrected water vapor signal we subtract the two signals

$$N_{WV} = N_{407} - N_{410}$$

Since both signals follow Poisson statistics and we can assume that there are uncorrelated, the standard deviation of the corrected signal is

$$std(N_{WV}) = \sqrt{N_{407}} + \sqrt{N_{410}} = \sqrt{N_{WV} + N_F} + \sqrt{N_F}$$

While the uncorrected signal is

$$std(N_{407}) = \sqrt{N_{WV} + N_F}$$

Since the fluorescence and water vapor signal have similar magnitude N, we can compare both as

$$\sqrt{2N} + \sqrt{N} > \sqrt{2N}$$

Which is similar to what is observed in Fig. 6c (i.e. ~70 percent increase in the random uncertainty).

**Ln.438. "but k is substantially more difficult to quantify and much more variable as it depends on the composition of the interfering aerosols" Why does this coefficient depends on aerosol type? I think only absolute value of fluorescence signal depends, but not kδ••[1].**

$k_F^{407 \rightarrow 410}$ is basically the ratio of the spectral intensity at 407 and 410 nm. Since the spectral emission shape of the aerosol fluorescence depends on the composition and type of the interfering aerosol, $k_F^{407 \rightarrow 410}$ depends of the aerosol composition, which is generally unknown. Dr. Reichardt's comment shows an example of this and estimates that the error of assuming $k_F^{407 \rightarrow 410} = 1$ is around 8% for that particular aerosol type.

**Comments by Dr. Jens Reichardt**

**(1) The authors had to expend great effort to establish the working hypothesis that fluorescence by stratospheric BBA caused the wet bias. Monthly averages and satellite aerosol data seem to be sufficient, but how much easier would the study have been if some simultaneous lidar aerosol data had been available. All lidar described are extremely potent instruments, so it should be relatively easy to augment the lidar data set with aerosol parameters, if budgets permit.**

While all of the instruments used in this study already have some aerosol characterization capabilities, their coverage, processing, and data availability strongly differ. We considered that using a common satellite dataset was sufficient for this particular study, which used this information mainly in a 'qualitative' way. Nevertheless, we certainly agree with the need to enhance the aerosol capabilities of these lidars, including a better coverage in the UTLS at our JPL Table Mountain Facility as well as a routine data archiving approach among NDACC sites.

**(2) The referee fully agrees with the assessment given concerning the shape of the BBA fluorescence spectrum and the potential use of a spectrometer (or at least two discrete detection channels on both sides of the main water vapor Raman line) to account for spectral gradients around 407 nm. Measurements with the spectrometric RAMSES lidar reveal just that in BBA filaments at the tropopause (see figure). At 11.5 km (red curve), the spectral backscatter coefficient at 407.5 nm is 7%-8% smaller than at 410.5 nm. So in this case the current correction scheme would have introduced a dry bias to the mixing-ratio results.**

The figure included is indeed very illustrative of the challenges associated with the correction of fluorescence and the limitations of the proposed method. While the use of a spectrometer is certainly an option, some issues like fiber fluorescence (if fiber coupling is used), channel cross-talk, and generally stronger background due to stray light are a concern for this type of instruments aiming to the UTLS. Also, additional signals included as part of the correction likely mean a larger random uncertainty. Using two filters with the same center wavelength (407 nm) with different width (e.g. 0.5 nm and 1 nm will both have basically the same amount of WV signal but the 1 nm will be affected by fluorescence twice as

much) might be the best solution as it minimizes the impact of the fluorescence spectral shape without compromising the performance. Also, just changing the transmission wavelength to 308 nm or 532 nm might be an overall better solution.

**Specific comments:**

**l. 248: Given the scarcity of stratospheric measurements a slightly more defensive wording would be warranted. Probably, the reason why there are data points at all is that the additional fluorescence signal pushes the statistical errors of apparent water vapor mixing ratio below the error threshold.**

We modified this sentence and indicated that this corresponds to 'apparent' increase in water vapor. We added the following sentence to reflect this: *"In fact, it is this enhancement of the water vapor signal by fluorescence what pushes the random uncertainty of the retrieval under the cutoff threshold."*

**l. 253: Here the fluorescence is missing, hence, no (additional) signal in the water-vapor detection channel and thus the statistical errors remain high and above the error threshold. Maybe this should be mentioned.**

We believe that the previously added sentence indeed clarifies this point (additional photons by fluorescence push the signal above the uncertainty threshold).

**ls. 293, 301: See 'general comments' above. If fluorescence increases with wavelength, the corrected water vapor mixing ratio will be too low.**

The following sentence was added to reflect this issue *"Any deviation from this assumption will translate in a bias in the corrected water vapor. If fluorescence increases with wavelength, the correction will introduce a dry bias, while the opposite will happen in case of a fluorescence spectrum that decreases with wavelength."*

**Fig. 6: It is quite amazing that the aerosol effect is seen in the entire lower stratosphere up to almost 20 km. Did BBA get up that high in California? For instance, in northern Germany the maximum height of BBA detection reached ~16 km only once in 2020 and was always below ~13 km in 2021. A similar plot for a time period without stratospheric BBA contamination would probably be worthwhile.**

An enhancement of the aerosol backscatter ratio is visible for 2020 and 2021 in Fig. 1a up to around 20 km asl. The general difference between the plume elevation observed at the two sites (Table Mountain vs Lindenberg) is likely mainly due to the latitude difference and associated tropopause height change.

**Further, mixing ratios in the upper troposphere over California are already so small that significant BBA fluorescence should interfere here with the humidity measurement as well. The reviewer observes this phenomenon in Germany quite often. Do you have such measurement cases?**

We haven't specifically looked at fluorescence in the upper troposphere since the fluorescence monitoring channel was installed on TMWAL, but the proposed approach should make no difference between LS and UT. Eventually, if we get a clear tropospheric transport event we will take a close look at it.

**Finally, is the RS41 data product in the stratosphere actually good enough for this comparison?**

Generally speaking, the RS41s seem to be in good agreement with the CFHs up to around 20 km, which should be sufficient for this study (see example below, RS41 vs CFH vs iMet).

[Figure]

**Technical corrections:**

**l. 58: Aerosol typing is also discussed in: Reichardt, J., R. Leinweber, and A. Schwebe, 2018: Fluorescing aerosols and clouds: Investigations of co-existence. EPJ Web Conf., 176, 05010, https://doi.org/10.1051/epjconf/201817605010.**

We included the suggested reference.

**l. 171: … product from…**

Corrected.

**l. 189: trustworthily**

Corrected.

**l. 200: … of the with… (Something is missing here.)**

The complete sentence should read as follows: *"However, the calibration was also based on the comparison of the integration of the lidar water vapor profile with collocated global navigation satellite system integrated water vapor measurements."*

**l. 330 ff.: Subscripts that are not variables must be in normal text style, not italic.**

Equations have been changed to reflect this.

**l. 343: See 'general comments' above.**

This was addressed in l. 295.

**Fig. 5c), l. 393: Is the range 6-8 km or 9-11 km?5**

9-11 km is the correct range. Figure 5 caption was modified to reflect this change.

[revised manuscript text omitted]
^{407\to410}_{L} \cdot (k^{407\to410}_{WV} \cdot N^{407}_{WV}(R) + k^{407\to410}_{F} \cdot N^{407}_{F}(R) + k^{407\to410}_{B} \cdot N^{407}_{BKG})$$
$$(2)$$

where $k^{407\to410}_{WV}$ , $k^{407\to410}_{F}$ and $k^{407\to410}_{B}$ are proportionality constants given by the spectral
characteristics of the water vapor Raman, fluorescence and background signal, respectively. As mentioned before, we assume
that the fluorescence spectra of the aerosols affecting the water vapor measurements can be considered to be constant over the
3 nm that separate the water vapor and the fluorescence monitoring channels, making $k^{407\to410}_{F} = 1$. On the other
hand, $k^{407\to410}_{L}$ is the lidar spectral response ratio and corresponds to the change in the lidar efficiency between these
two wavelengths due to differences in filter transmission, filter width, PMT efficiency, etc. If we pick a range $R_{BKG}$ where no
water vapor or fluorescence contribution is expected ($N^{407}_{WV}(R_{BKG}) = 0$ and $N^{407}_{F}(R_{BKG}) = 0$), (1) and (2) can
be written as:

$$N^{407}(R_{BKG}) = N^{407}_{BKG}$$
$$(3)$$

$$N^{410}(\text{}R_{\text{BKG}}) = \text{}k_L^{407\to410} \cdot \text{}k_B^{407\to410} \cdot \text{}N_{\text{BKG}}^{407}$$

(4)

360

Since the wavelength difference between the $N^{407}$ and $N^{410}$ channels are small, the background radiation at these two wavelengths are expected to be very close, and thus $k_B^{407\to410} = 1$. In order to check this assumption, we retrieved the top of the atmosphere (TOA) moonlight spectra based on the algorithm provided by Miller and Turner (2009). The 410/407 nm spectral ratio at the TOA is 1.03, and mostly independent from the moon phase. If we then assume that the atmospheric

365  extinction is dominated by Rayleigh scattering around 400 nm (Cramer et al., 2013) and use a direct viewing geometry, we obtain $= k_B^{407\to410} = 1.03 \cdot \left(\frac{407\ \text{nm}}{410\ nm}\right)^4 = 1$. While the direct moon viewing does not correspond to a real observation geometry, we expect the deviations from this assumption to be small based on other scattered moonlight model results (Jones et al., 2013).

370  Based on this assumption, we can derive our lidar spectral response ratio as

$$\text{}k_L^{407\to410} = \frac{N^{410}(R_{BKG})}{N^{407}(R_{BKG})}$$

(5)

375  Because $k_L^{407\to410}$ is related to the characteristics of the lidar components, we expect it to remain mostly constant over time, as long as there is no change in the lidar setup. Also, as mentioned before, this calculation assumes that the background signals are dominated by moonlight sky background and that the contribution of the PMT dark counts and other light sources with unknown spectral characteristics is negligible. In order to verify this assumption, we analyzed the stability of the ratio defined in Eq. 5 as a function of the background levels.

[Figure]

**Figure 5.** (a) Time series of background levels for the 407 nm (dots, red) and 410 nm (dots, black) channels. The average of the background levels for days where other unknown light sources dominate (low moonlight contribution) is also shown (dashed). (b) Time series of the 410/407 nm channel background ratio (lidar spectral response ratio) calculated based on days where moonlight dominates and after subtracting the contribution from unknown light sources. The average of the lidar spectral response ratio is also shown (dashed, black). (c) Time series of the ratio between the 410/407 nm channels for altitudes between 9-11 km a.s.l., where the fluorescence contribution can be typically considered negligible (Eq. 6). The average is also shown (dashed, black).

The background levels presented in Fig. 5a show a clear temporal pattern caused by the moon phases, with a maximum background occurring during the full moon phase. As the moon illumination drops towards new moon, we can see that the background levels stabilize to relatively constant values, while maintaining a substantial difference between the 407 nm and 410 nm channels. Since these two channels share the same PMT and optical path (with the exception of a changing filter), we can conclude that these background levels are not related to PMT dark counts but rather to a different background source (i.e. light pollution from Los Angeles basin and other nearby sources). Since the calculation of $k_L^{407 \to 410}$ assumes the knowledge of the background spectral characteristics, we first estimated the contribution of the light pollution as the average of the background for days where the moonlight contribution is negligible (Fig. 5a). This average was then subtracted from the background levels of moon dominated days in order to keep only the moonlight contribution, that has a known spectrum. The ratio of these two channels after correction (Fig. 5b), exhibit a fairly constant value, with an average of $3.8 \pm 0.08$.

Next, the constant term $\cancel{k_{WV}^{407\rightarrow410}} k_{WV}^{407\rightarrow410}$ can be determined by calculating the ratio of the 407 nm and 410 nm signals (after background subtraction) for a range R at which the fluorescence contribution is negligible, or else using laboratory

400 measurements of the water vapor Raman spectrum. From Eqs. 1 and 2, after background subtraction (denoted by *), and assuming $\cancel{N_F^{407}} N_F^{407}(R) \ll N_{\cancel{WV}\,WV}^{407}(R)$

$$k_{\cancel{L}}^{407\rightarrow410} \cdot k_{\cancel{WV}}^{407\rightarrow410} k_L^{407\rightarrow410} \cdot k_{WV}^{407\rightarrow410} = \frac{N^{410*}(R)}{N^{407*}(R)}$$

(6)

405

Figure 5c shows the result of applying Eq. 6 to each measurement period between 9-11 km a.s.l., where $\cancel{N_F^{407}} N_F^{407}(R) \ll$ $\cancel{N_{WV}^{407}} N_{WV}^{407}(R)$. This ratio shows a relatively stable behavior and a mean value of 0.16±0.03, which translates into $\cancel{k_{WV}^{407\rightarrow410}=}$ $k_{WV}^{407\rightarrow410} = 0.042$. This result is relatively close to the one obtained by the other proposed method (laboratory measurements of the water vapor Raman cross-section) and shown in Fig. 4 as the two black dots with a ratio of about 0.1. The difference

410 between these two approaches can be attributed to several different factors not accounted for in the calculation, like departure of the filter transfer functions from the values used for the calculation, changes in the PMT response function between the two wavelengths, etc.

After background subtraction we have the following equation system, where we can obtain the fluorescence-corrected water

415 vapor signal needed to conduct the standard Raman lidar water vapor retrieval:

$$N^{407*}(R) = \cancel{N_{WV}^{407}} N_{WV}^{407}(R) + N_{\cancel{F}\,F}^{407}(R)$$

(7)

420 $$N^{410*}(R) = k_{\cancel{L}}^{407\rightarrow410} k_L^{407\rightarrow410} \cdot k_{\cancel{WV}}^{407\rightarrow410} k_{WV}^{407\rightarrow410} \cdot \cancel{N_{WV}^{407}} N_{WV}^{407}(R) + k_{\cancel{L}}^{407\rightarrow410} k_L^{407\rightarrow410} \cdot \cancel{N_F^{407}} N_F^{407}(R)$$

(8)

From Eq. 7

425 $$\cancel{N_{WV}^{407}} N_{WV}^{407}(R) = N^{407*}(R) - \cancel{N_F^{407}} N_F^{407}(R)$$

(9)

While from Eq. 8

$$\cancel{N_F^{407}}N_F^{407}(R) = \frac{1}{\cancel{k_L^{407\to410}}} \frac{1}{k_L^{407\to410}} N^{410*}(R) - \cancel{k_{WV}^{407\to410}}k_{WV}^{407\to410} \cdot \cancel{N_{WV}^{407}}N_{WV}^{407}(R)$$

$$(10)$$

[revised manuscript text omitted]

---

## Author Response (AR2)

Dear reviewers and editor

We would like to thank you for valuable comments on our manuscript. The original comments are in bold, followed by our answers and modifications introduced to the manuscript.

**Comments by Anonymous Referee #1**

**Fig.1. Green letters M-09 at the top of the figure probably mean MOHAVE-2009. This should be explained in the capture.**

We added '(green)' as a clarification to Fig. 1a caption:

"Major eruptions (black), PyroCb events (red), and the MOHAVE-2009 *(green)* field-campaign are indicated at the top of the plot (see Table 1)."

**Influence of fluorescence on the vapor measurements discussed also in: Reichardt, J.: Cloud and aerosol spectroscopy with Raman lidar, J. Atm. Ocean. Tech., 31, 1946-1963, 2014.**

We added a reference to that work as part of the following sentence:

"The first report of aerosol fluorescence contamination on a water vapor Raman lidar measurement was presented by Immler et al. (2005a) and further discussed by Immler et al. (2005b) and *Reichardt (2014)*."

**Fig.5c. From the title of left axis it is not clear, that this is the ratio of the channels**

Since Fig. 5c is the ratio of the channels after background subtraction we prefer to leave the notation as is. A reference to Eq. 6 explaining the notation is included in the figure caption.

**Fig.6c. I am a bit confused by the explanation of increase of uncertainty at high altitudes after correction. If we measure the sum of water vapor and fluorescence signal, does it mean that statistical uncertainty of vapor measurements becomes lower? I think uncertainty of vapor signal should stay the same…Correct me if I am wrong.**

If we simplify the problem (e.g. the instrumental response is the same for the WV and fluorescence channel, there is no background or cross-talk), the signal measured by the WV channel ($N_{407}$) is the sum of the water vapor Raman signal ($N_{WV}$) and the fluorescence signal $N_F$.

$$N_{407} = N_{WV} + N_F$$

While in the fluorescence monitoring channel ($N_{410}$)

$$N_{410} = N_F$$

To calculated the corrected water vapor signal we subtract the two signals

$$N_{WV} = N_{407} - N_{410}$$

Since both signals follow Poisson statistics and we can assume that there are uncorrelated, the standard deviation of the corrected signal is

$$std(N_{WV}) = \sqrt{N_{407}} + \sqrt{N_{410}} = \sqrt{N_{WV} + N_F} + \sqrt{N_F}$$

While the uncorrected signal is

$$std(N_{407}) = \sqrt{N_{\mathrm{WV}} + N_{\mathrm{F}}}$$

Since the fluorescence and water vapor signal have similar magnitude N, we can compare both as

$$\sqrt{2N} + \sqrt{N} > \sqrt{2N}$$

Which is similar to what is observed in Fig. 6c (i.e. ~70 percent increase in the random uncertainty).

**Ln.438. "but k is substantially more difficult to quantify and much more variable as it depends on the composition of the interfering aerosols" Why does this coefficient depends on aerosol type? I think only absolute value of fluorescence signal depends, but not $k_{\mathrm{F}}^{\bullet\bullet}$[1].**

$k_{\mathrm{F}}^{407\to410}$ is basically the ratio of the spectral intensity at 407 and 410 nm. Since the spectral emission shape of the aerosol fluorescence depends on the composition and type of the interfering aerosol, $k_{\mathrm{F}}^{407\to410}$ depends of the aerosol composition, which is generally unknown. Dr. Reichardt's comment shows an example of this and estimates that the error of assuming $k_{\mathrm{F}}^{407\to410} = 1$ is around 8% for that particular aerosol type.

**Comments by Dr. Jens Reichardt**

**(1) The authors had to expend great effort to establish the working hypothesis that fluorescence by stratospheric BBA caused the wet bias. Monthly averages and satellite aerosol data seem to be sufficient, but how much easier would the study have been if some simultaneous lidar aerosol data had been available. All lidar described are extremely potent instruments, so it should be relatively easy to augment the lidar data set with aerosol parameters, if budgets permit.**

While all of the instruments used in this study already have some aerosol characterization capabilities, their coverage, processing, and data availability strongly differ. We considered that using a common satellite dataset was sufficient for this particular study, which used this information mainly in a 'qualitative' way. Nevertheless, we certainly agree with the need to enhance the aerosol capabilities of these lidars, including a better coverage in the UTLS at our JPL Table Mountain Facility as well as a routine data archiving approach among NDACC sites.

**(2) The referee fully agrees with the assessment given concerning the shape of the BBA fluorescence spectrum and the potential use of a spectrometer (or at least two discrete detection channels on both sides of the main water vapor Raman line) to account for spectral gradients around 407 nm. Measurements with the spectrometric RAMSES lidar reveal just that in BBA filaments at the tropopause (see figure). At 11.5 km (red curve), the spectral backscatter coefficient at 407.5 nm is 7%-8% smaller than at 410.5 nm. So in this case the current correction scheme would have introduced a dry bias to the mixing-ratio results.**

The figure included is indeed very illustrative of the challenges associated with the correction of fluorescence and the limitations of the proposed method. While the use of a spectrometer is certainly an option, some issues like fiber fluorescence (if fiber coupling is used), channel cross-talk, and generally stronger background due to stray light are a concern for this type of instruments aiming to the UTLS. Also, additional signals included as part of the correction likely mean a larger random uncertainty. Using two filters with the same center wavelength (407 nm) with different width (e.g. 0.5 nm and 1 nm will both have basically the same amount of WV signal but the 1 nm will be affected by fluorescence twice as

much) might be the best solution as it minimizes the impact of the fluorescence spectral shape without compromising the performance. Also, just changing the transmission wavelength to 308 nm or 532 nm might be an overall better solution.

**Specific comments:**

**l. 248: Given the scarcity of stratospheric measurements a slightly more defensive wording would be warranted. Probably, the reason why there are data points at all is that the additional fluorescence signal pushes the statistical errors of apparent water vapor mixing ratio below the error threshold.**

We modified this sentence and indicated that this corresponds to 'apparent' increase in water vapor. We added the following sentence to reflect this: *"In fact, it is this enhancement of the water vapor signal by fluorescence what pushes the random uncertainty of the retrieval under the cutoff threshold."*

**l. 253: Here the fluorescence is missing, hence, no (additional) signal in the water-vapor detection channel and thus the statistical errors remain high and above the error threshold. Maybe this should be mentioned.**

We believe that the previously added sentence indeed clarifies this point (additional photons by fluorescence push the signal above the uncertainty threshold).

**ls. 293, 301: See 'general comments' above. If fluorescence increases with wavelength, the corrected water vapor mixing ratio will be too low.**

The following sentence was added to reflect this issue *"Any deviation from this assumption will translate in a bias in the corrected water vapor. If fluorescence increases with wavelength, the correction will introduce a dry bias, while the opposite will happen in case of a fluorescence spectrum that decreases with wavelength."*

**Fig. 6: It is quite amazing that the aerosol effect is seen in the entire lower stratosphere up to almost 20 km. Did BBA get up that high in California? For instance, in northern Germany the maximum height of BBA detection reached ~16 km only once in 2020 and was always below ~13 km in 2021. A similar plot for a time period without stratospheric BBA contamination would probably be worthwhile.**

An enhancement of the aerosol backscatter ratio is visible for 2020 and 2021 in Fig. 1a up to around 20 km asl. The general difference between the plume elevation observed at the two sites (Table Mountain vs Lindenberg) is likely mainly due to the latitude difference and associated tropopause height change.

**Further, mixing ratios in the upper troposphere over California are already so small that significant BBA fluorescence should interfere here with the humidity measurement as well. The reviewer observes this phenomenon in Germany quite often. Do you have such measurement cases?**

We haven't specifically looked at fluorescence in the upper troposphere since the fluorescence monitoring channel was installed on TMWAL, but the proposed approach should make no difference between LS and UT. Eventually, if we get a clear tropospheric transport event we will take a close look at it.

**Finally, is the RS41 data product in the stratosphere actually good enough for this comparison?**

Generally speaking, the RS41s seem to be in good agreement with the CFHs up to around 20 km, which should be sufficient for this study (see example below, RS41 vs CFH vs iMet).

[Figure]

**Technical corrections:**

**l. 58: Aerosol typing is also discussed in: Reichardt, J., R. Leinweber, and A. Schwebe, 2018: Fluorescing aerosols and clouds: Investigations of co-existence. EPJ Web Conf., 176, 05010, https://doi.org/10.1051/epjconf/201817605010.**

We included the suggested reference.

**l. 171: … product from…**

Corrected.

**l. 189: trustworthily**

Corrected.

**l. 200: … of the with… (Something is missing here.)**

The complete sentence should read as follows: *"However, the calibration was also based on the comparison of the integration of the lidar water vapor profile with collocated global navigation satellite system integrated water vapor measurements."*

**l. 330 ff.: Subscripts that are not variables must be in normal text style, not italic.**

Equations have been changed to reflect this.

**l. 343: See 'general comments' above.**

This was addressed in l. 295.

**Fig. 5c), l. 393: Is the range 6-8 km or 9-11 km?5**

9-11 km is the correct range. Figure 5 caption was modified to reflect this change.

**Comments by Dr. Ulla Wandinger (Editor)**

**Regarding the comment of Dr. Reichardt on Fig. 6 (height of BBA layers) and your discussion of a new quality of fire events since 2017 in Sec. 5 (line 476 ff.), you may want to consider more information avaible in the literature. The following publications (and references therein) provide inside into the presence of smoke aerosol in the stratosphere of the northern and southern hemisphere since 2017 and on self-lofting processes that lead to the rise of layers up to more than 30 km height.**

Thank you for pointing out these additional references. There is indeed abundant literature, specially with regards to the 2017 Canadian PyroCb, 2019 Siberian wildfires and 2020 Australian wildfires. The following references were added to the manuscript:

P9L239 "… occurring in western North America on 12 August 2017 (Peterson et al., 2018; *Baars et al., 2019; Ansmann et al., 2018*)."

P19L468 *"It has been shown that the smoke plumes originated from these fires are subject to self-lofting which enhance their vertical development over time (Ohneiser et al., 2022)."*

In the case of the 2020-2021 northern hemisphere BBA plume mentioned by Dr. Reichardt, it is still TBD which specific fire(s) were responsible for the observed enhancement, but the difference in the maximum height observed between the two sites is likely mainly linked to the difference in latitude between Lindenberg (52 N) and Table Mountain Facility (34 N).

[revised manuscript text omitted]

(1)

$\cancel{N_{WV}^{407}} N_{WV}^{407}$ represents the contribution from water vapor Raman scattering, $\cancel{N_{F}^{407}} N_{F}^{407}$ represents fluorescence, and $\cancel{N_{BKG}^{407}} N_{BKG}^{407}$

340 represents the background contribution (dominated by moonlight and sky background rather than PMT dark counts).

Similarly, the number of photons measured on the new fluorescence channel $N^{410}$ at the same range gate can be written in terms of the same components as

$$N^{410}(R) = \cancel{k_{L}^{407 \to 410} \cdot (k_{WV}^{407 \to 410} \cdot N_{WV}^{407}(R) + k_{F}^{407 \to 410} \cdot N_{F}^{407}(R) + k_{B}^{407 \to 410} \cdot N_{BKG}^{407})} k_{L}^{407 \to 410} \cdot (k_{WV}^{407 \to 410} \cdot N_{WV}^{407}(R) +$$

345 $$k_{F}^{407 \to 410} \cdot N_{F}^{407}(R) + k_{B}^{407 \to 410} \cdot N_{BKG}^{407})$$ (2)

where $\cancel{k_{WV}^{407 \to 410}} k_{WV}^{407 \to 410}$, $\cancel{k_{F}^{407 \to 410}} k_{F}^{407 \to 410}$ and $\cancel{k_{B}^{407 \to 410}} k_{B}^{407 \to 410}$ are proportionality constants given by the spectral characteristics of the water vapor Raman, fluorescence and background signal, respectively. As mentioned before, we assume that the fluorescence spectra of the aerosols affecting the water vapor measurements can be considered to be constant over the

350 3 nm that separate the water vapor and the fluorescence monitoring channels, making $\cancel{k_{F}^{407 \to 410}} k_{F}^{407 \to 410} = 1$. On the other hand, $\cancel{k_{L}^{407 \to 410}} k_{L}^{407 \to 410}$ is the lidar spectral response ratio and corresponds to the change in the lidar efficiency between these two wavelengths due to differences in filter transmission, filter width, PMT efficiency, etc. If we pick a range $R_{BKG}$ where no water vapor or fluorescence contribution is expected ($\cancel{N_{WV}^{407}} N_{WV}^{407}(R_{BKG}) = 0$ and $\cancel{N_{F}^{407}} N_{F}^{407}(\cancel{R_{BKG}} R_{BKG}) = 0$), (1) and (2) can be written as:

355

$$N^{407}(\cancel{R_{BKG}} R_{BKG}) = \cancel{N_{BKG}^{407}} N_{BKG}^{407}$$

(3)

$$N^{410}(R_{\textcolor{red}{\sout{BKG}}},R_{\text{BKG}}) = \textcolor{red}{\sout{k_L^{407\to410}}}k_L^{407\to410} \cdot \textcolor{red}{\sout{k_B^{407\to410}}}k_B^{407\to410} \cdot \textcolor{red}{\sout{N_{BKG}^{407}}}N_{\text{BKG}}^{407}$$

$$(4)$$

Since the wavelength difference between the $N^{407}$ and $N^{410}$ channels are small, the background radiation at these two wavelengths are expected to be very close, and thus $\textcolor{red}{\sout{k_B^{407\to410}}}k_B^{407\to410} = 1$. In order to check this assumption, we retrieved the top of the atmosphere (TOA) moonlight spectra based on the algorithm provided by Miller and Turner (2009). The 410/407 nm spectral ratio at the TOA is 1.03, and mostly independent from the moon phase. If we then assume that the atmospheric extinction is dominated by Rayleigh scattering around 400 nm (Cramer et al., 2013) and use a direct viewing geometry, we obtain $\textcolor{red}{\sout{k_B^{407\to410}}} = k_B^{407\to410} = 1.03 \cdot \left(\frac{407\,\text{nm}}{410\,nm}\right)^4 = 1$. While the direct moon viewing does not correspond to a real observation geometry, we expect the deviations from this assumption to be small based on other scattered moonlight model results (Jones et al., 2013).

Based on this assumption, we can derive our lidar spectral response ratio as

$$\textcolor{red}{\sout{k_L^{407\to410}}}k_L^{407\to410} = \frac{N^{410}(R_{BKG})}{N^{407}(R_{BKG})}$$

$$(5)$$

Because $\textcolor{red}{\sout{k_L^{407\to410}}}k_L^{407\to410}$ is related to the characteristics of the lidar components, we expect it to remain mostly constant over time, as long as there is no change in the lidar setup. Also, as mentioned before, this calculation assumes that the background signals are dominated by moonlight sky background and that the contribution of the PMT dark counts and other light sources with unknown spectral characteristics is negligible. In order to verify this assumption, we analyzed the stability of the ratio defined in Eq. 5 as a function of the background levels.

[Figure]

**Figure 5.** (a) Time series of background levels for the 407 nm (dots, red) and 410 nm (dots, black) channels. The average of the background levels for days where other unknown light sources dominate (low moonlight contribution) is also shown (dashed). (b) Time series of the 410/407 nm channel background ratio (lidar spectral response ratio) calculated based on days where moonlight dominates and after subtracting the contribution from unknown light sources. The average of the lidar spectral response ratio is also shown (dashed, black). (c) Time series of the ratio between the 410/407 nm channels for altitudes between 9-11 km a.s.l., where the fluorescence contribution can be typically considered negligible (Eq. 6). The average is also shown (dashed, black).

The background levels presented in Fig. 5a show a clear temporal pattern caused by the moon phases, with a maximum background occurring during the full moon phase. As the moon illumination drops towards new moon, we can see that the background levels stabilize to relatively constant values, while maintaining a substantial difference between the 407 nm and 410 nm channels. Since these two channels share the same PMT and optical path (with the exception of a changing filter), we can conclude that these background levels are not related to PMT dark counts but rather to a different background source (i.e. light pollution from Los Angeles basin and other nearby sources). Since the calculation of $k_L^{407\rightarrow410}$ assumes the knowledge of the background spectral characteristics, we first estimated the contribution of the light pollution as the average of the background for days where the moonlight contribution is negligible (Fig. 5a). This average was then subtracted from the background levels of moon dominated days in order to keep only the moonlight contribution, that has a known spectrum. The ratio of these two channels after correction (Fig. 5b), exhibit a fairly constant value, with an average of 3.8±0.08.

Next, the constant term  $k_{WV}^{407\to410}$ can be determined by calculating the ratio of the 407 nm and 410 nm signals (after background subtraction) for a range R at which the fluorescence contribution is negligible, or else using laboratory measurements of the water vapor Raman spectrum. From Eqs. 1 and 2, after background subtraction (denoted by *), and assuming $N_F^{407}(R) \ll$ $N_{WV}^{407}(R)$

$$\text{}k_{WV}^{407\to410}\text{~~}k_L^{407\to410} \cdot k_{WV}^{407\to410} = \frac{N^{410*}(R)}{N^{407*}(R)}$$

(6)

Figure 5c shows the result of applying Eq. 6 to each measurement period between 9-11 km a.s.l., where $N_F^{407}(R) \ll$ $N_{WV}^{407}(R)$. This ratio shows a relatively stable behavior and a mean value of 0.16±0.03, which translates into  $k_{WV}^{407\to410} = 0.042$. This result is relatively close to the one obtained by the other proposed method (laboratory measurements of the water vapor Raman cross-section) and shown in Fig. 4 as the two black dots with a ratio of about 0.1. The difference between these two approaches can be attributed to several different factors not accounted for in the calculation, like departure of the filter transfer functions from the values used for the calculation, changes in the PMT response function between the two wavelengths, etc.

After background subtraction we have the following equation system, where we can obtain the fluorescence-corrected water vapor signal needed to conduct the standard Raman lidar water vapor retrieval:

$$N^{407*}(R) = \text{}N_{WV}^{407}(R) + N_{F}^{407}(R)$$

(7)

$$N^{410*}(R) = \text{}k_L^{407\to410} \cdot \text{}k_{WV}^{407\to410} \cdot \text{}N_{WV}^{407}(R) + \text{}k_L^{407\to410} \cdot \text{}N_F^{407}(R)$$

(8)

From Eq. 7

$$\text{}N_{WV}^{407}(R) = N^{407*}(R) - \text{}N_F^{407}(R)$$

(9)

[revised manuscript text omitted]